# Esrrb extinction triggers dismantling of naïve pluripotency and marks commitment to differentiation

Nicola Festuccia[1,†,*] (iD), Florian Halbritter[1,‡] (iD), Andrea Corsinotti[1,2], Alessia Gagliardi[1,§], Douglas Colby[1], Simon R Tomlinson[1] & Ian Chambers[1,**] (iD)

## Abstract

Self-renewal of embryonic stem cells (ESCs) cultured in LIF/fetal calf serum (FCS) is incomplete with some cells initiating differentiation. While this is reflected in heterogeneous expression of naive pluripotency transcription factors (TFs), the link between TF heterogeneity and differentiation is not fully understood. Here, we purify ESCs with distinct TF expression levels from LIF/FCS cultures to uncover early events during commitment from naïve pluripotency. ESCs carrying fluorescent *Nanog* and *Esrrb* reporters show Esrrb downregulation only in Nanog[low] cells. Independent *Esrrb* reporter lines demonstrate that Esrrb[negative] ESCs cannot effectively self-renew. Upon Esrrb loss, pre-implantation pluripotency gene expression collapses. ChIP-Seq identifies different regulatory element classes that bind both OCT4 and NANOG in Esrrb[positive] cells. Class I elements lose NANOG and OCT4 binding in Esrrb[negative] ESCs and associate with genes expressed preferentially in naïve ESCs. In contrast, Class II elements retain OCT4 but not NANOG binding in ESRRB-negative cells and associate with more broadly expressed genes. Therefore, mechanistic differences in TF function act cumulatively to restrict potency during exit from naïve pluripotency.

**Keywords** commitment; embryonic stem cells; fluorescent reporters; self-renewal; transcription factors

**Subject Categories** Development & Differentiation; Stem Cells; Transcription

The EMBO Journal (2018) 37: e95476

## Introduction

Naïve pluripotency is a characteristic of pre-implantation mouse epiblast cells and their *in vitro* derivatives, embryonic stem cells (ESCs). The activity of a relatively well-characterized pluripotency gene regulatory network, centered on the triumvirate OCT4, SOX2 and NANOG, controls the dual abilities of ESCs to self-renew and to differentiate (Jaenisch & Young, 2008; Chambers & Tomlinson, 2009; Ng & Surani, 2011). While populations of ESCs cultured in LIF/FCS express relatively homogeneous levels of Oct4 and Sox2, individual ESCs show varying levels of Nanog expression (Chambers *et al*, 2007; Kalmar *et al*, 2009; Abranches *et al*, 2014). Heterogeneous expression has also been reported for Esrrb (van den Berg *et al*, 2008), Klf4 (Niwa *et al*, 2009), Tbx3 (Niwa *et al*, 2009; Russell *et al*, 2015), Rex1 (Toyooka *et al*, 2008), and Stella (Hayashi *et al*, 2008). This heterogeneity is a manifestation of the simultaneous possession of differentiation and self-renewal potential within the ESC population (reviewed in Torres-Padilla & Chambers, 2014). Indeed, ESCs cultured in LIF/FCS have been proposed to contain a mixture of pluripotent cells that have progressed to different extents toward a primed state (Hackett & Surani, 2014). Such a primed state can also be captured *in vitro* as EpiSC, by explantation and culture of post-implantation epiblast cells (Brons *et al*, 2007; Tesar *et al*, 2007). In contrast, ESCs cultured in LIF supplemented with inhibitors of both MEK and GSK3b (two inhibitors or 2i; Ying *et al*, 2008) may represent the opposite end of the range of pluripotent states that can be maintained *in vitro* (Hackett & Surani, 2014). Supporting the idea that heterogeneity is a direct physiological consequence of the balance between self-renewal and differentiation, TF expression becomes uniformly high as a result of pharmacological treatment with PD0325901 (Wray *et al*, 2010). This MEK inhibitor blocks ESC differentiation by preventing the pro-differentiative action of autocrine FGF4 (Kunath *et al*, 2007; Stavridis *et al*, 2007), a direct target gene of OCT4 and SOX2 (Ambrosetti *et al*, 1997). Therefore, fractionating ESC populations cultured in LIF/FCS on the basis of heterogeneous TF expression has the potential to clarify mechanisms underlying the ability of ESCs to respond to

1  MRC Centre for Regenerative Medicine, Institute for Stem Cell Research, School of Biological Sciences, University of Edinburgh, Edinburgh, UK
2  Department of Anatomy and Embryology, Faculty of Medicine, University of Tsukuba, Ibaraki, Japan
   *Corresponding author. Tel: +44 20 8383 8228; E-mail: n.festuccia@lms.mrc.ac.uk
   **Corresponding author. Tel: +44 131 651 9562; Fax: +44 131 651 9501; E-mail: ichambers@ed.ac.uk
   †Present address: MRC London Institute of Medical Sciences, Institute of Clinical Sciences, Faculty of Medicine, Imperial College London, London, UK
   ‡Present address: Children's Cancer Research Institute, St. Anna Kinderkrebsforschung, Vienna, Austria
   §Present address: Canada's Michael Smith Genome Sciences Centre, BC Cancer, Vancouver, BC, Canada
   [Correction added on 9 October 2018, after first online publication: Affiliations have been corrected.]

autocrine signaling stimuli, dismantle the naïve pluripotency network, and exit self-renewal.

Previous studies have shown that NANOG controls the expression of a number of important pre-implantation pluripotency TFs (Festuccia *et al*, 2012) that are expressed differentially between ESCs and EpiLC/EpiSCs. Prominent among these is ESRRB, a TF that can confer cytokine-independent self-renewal when overexpressed and that can reprogram pluripotent cells from a primed to a naïve state (Festuccia *et al*, 2012). Here, we examine the effects of transcription factor heterogeneity in LIF/FCS. Using reporter lines for both Nanog and Esrrb expression, we FACS-purify subpopulations of ESCs expressing distinct Nanog/Esrrb levels. The effects of downregulation of these factors on pre-implantation pluripotency gene expression and the initial steps in dismantling of the naïve pluripotency network are then explored.

Expression of Esrrb is lost only in Nanog[negative] cells with loss of Esrrb coinciding with commitment to differentiation. Tracking downregulation of Esrrb expression enabled us to begin to identify the molecular changes accompanying extinction of the naïve pluripotency network. Genome-wide ChIP-Seq of sorted ESC populations identified two classes of regulatory elements active in naïve pluripotent cells: one in which OCT4 binding is dependent upon NANOG and ESRRB (Class I enhancers) and one in which OCT4 binding is independent of both NANOG and ESRRB (Class II enhancers). Importantly, Class I enhancers are specific to naïve pluripotent cells and lose OCT4 binding, becoming inactive in committed cells. In contrast, Class II enhancers are OCT4-bound and active in both ESC and EpiSC. This study therefore clarifies molecular events that drive the early stages of ESC differentiation.

# Results

To investigate the co-expression of NANOG and ESRRB at the single-cell level in LIF/FCS cultures, immunofluorescence was performed. This showed that cells expressing high levels of NANOG also tend to show high levels of ESRRB (Fig 1A). Quantitative analysis confirmed this and showed that cells in which NANOG was reduced below a threshold level exhibited a progressive loss of ESRRB expression. Above this threshold, a wide range of NANOG levels sustained high ESRRB expression. This resulted in the preferential detection of ESRRB[positive]/NANOG[low], but not NANOG[positive]/ESRRB[low] cells, prior to loss of both proteins (see curvature of the regression line, Fig 1B).

To further explore the relationship between expression of Esrrb and Nanog upon ESC phenotype, the coding sequence of the tdTomato red fluorescent protein was fused downstream of the *Esrrb* ORF by homologous recombination in wild-type E14Tg2a ESCs (Fig EV1, Appendix Fig S1A and B). Southern blot analysis identified clones carrying the correctly targeted structure at both 5′ and 3′ ends of *Esrrb* (Appendix Fig S1C and D). Examination of targeted Esrrb-tdTomato (E-tdT) cells showed that the Esrrb-tdTomato fusion protein had a similar half-life to that of endogenous Esrrb (Appendix Fig S2A and B). Immunofluorescence analysis of endogenous ESRRB and Esrrb-tdTomato protein expression confirmed that tdTomato was a reliable reporter of ESRRB expression (Appendix Fig S2C). Next, lines in which *Nanog* and *Esrrb* expression can be monitored by GFP and tdTomato, respectively, were generated by double targeting of the Esrrb-tdTomato fusion reporter in TNG cells (Chambers *et al*, 2007), in which a GFP-IRES-Puro[R] cassette is knocked-in to one of the *Nanog* alleles. Southern analysis (Appendix Fig S1C and D) identified correctly targeted TNG E-tdT ESCs (Fig EV1). Plating TNG E-tdT ESCs at clonal density produced colonies in which Nanog:GFP and Esrrb-tdTomato were expressed in centrally localized cells surrounded by a region of non-expressing cells (Fig 1C). Immunofluorescence and FACS analyses showed that, at this time, cells that have downregulated Esrrb-tdTomato continue to express OCT4, indicating that they remain pluripotent (Appendix Fig S2D). To examine molecular changes occurring early upon loss of Esrrb and Nanog, bulk cultures were next analysed. After passaging ESCs in LIF/FCS at high density, which minimizes heterogeneity, cells were replated at $2 \times 10^3$ cells/cm² and cultured for 3 days. FACS analysis of the undifferentiated SSEA-1-positive ESC population showed that GFP and tdTomato levels in double-positive cells were well correlated (Fig 1D). However, some cells in the population that had lost *Nanog*:GFP expression retained high Esrrb-tdTomato expression. In contrast, Esrrb-tdTomato[low] cells were not observed in the ESC populations expressing high Nanog: GFP. Continued application of puromycin, which selects for expression of *pac* from the *Nanog* locus, eliminated Esrrb-tdTomato[low] cells (Fig 1D). These findings indicate that heterogeneous *Esrrb* expression is confined to Nanog[negative] cells.

---

**Figure 1. NANOG acts upstream of *Esrrb* in ESCs.**

A, B Immunofluorescent detection of NANOG and ESRRB protein in wild-type E14Tg2a ESCs cultured for 3 days in LIF/FCS. (A) Widefield images of a representative colony. (B) Quantification of the mean NANOG and ESRRB fluorescence levels measured in nuclei identified by automatic segmentation of single optical plane images obtained by confocal microscopy. Values are expressed in arbitrary units (AU). OCT4[negative] cells, also identified by immunostaining, were excluded from the analysis. The red line represents the moving average of the data distribution. Representative of three independent experiments each including at least 1,000 nuclei.

C Colonies of TNG E-tdT ESCs showing GFP expression from *Nanog* and Esrrb-tdTomato fusion protein expression from *Esrrb* after 3 days in LIF/FCS.

D Flow cytometric analysis of Esrrb and Nanog fluorescent reporter expression in SSEA1[+] TNG E-tdT ESCs cultured for 3 days in LIF/FCS with or without puromycin.

E Immunofluorescent detection and quantification (as in panel B) of total NANOG and ESRRB protein in E14Tg2a ESCs overexpressing NANOG (left) or ESRRB (right) cultured for 3 days in LIF/FCS. Parental E14Tg2as are shown as a reference. The red lines indicate the negative thresholds defined by staining ESCs differentiated for 3 days in the absence of LIF and the presence of retinoic acid.

F Flow cytometric analysis of Esrrb-GFP and Nanog-mCherry fusion protein expression in SSEA1[+] NER ESCs cultured for 3 days in LIF/FCS.

G Histograms showing expression levels of Esrrb-GFP and Nanog-mCherry from the respective targeted endogenous alleles in SSEA1[+] NER ESCs overexpressing NANOG (left) or ESRRB (right) cultured for 3 days in LIF/FCS. Parental NER ESCs are shown as a reference, and wild-type E14Tg2a set the negative thresholds.

H Comparative flow cytometric analysis of *Esrrb*-2a-tdTomato and *Nanog*:GFP expression in undifferentiated SSEA-1[+] TNG E-2a-tdT or Nanog-null ESΔN E-2a-tdT ESCs cultured for 3 days in LIF/FCS. Data are shown as dot plots (top) and histogram (bottom).

Data information: For a schematic representation of the reporter allele configuration characteristic of each cell line, please refer to Fig EV1.

---

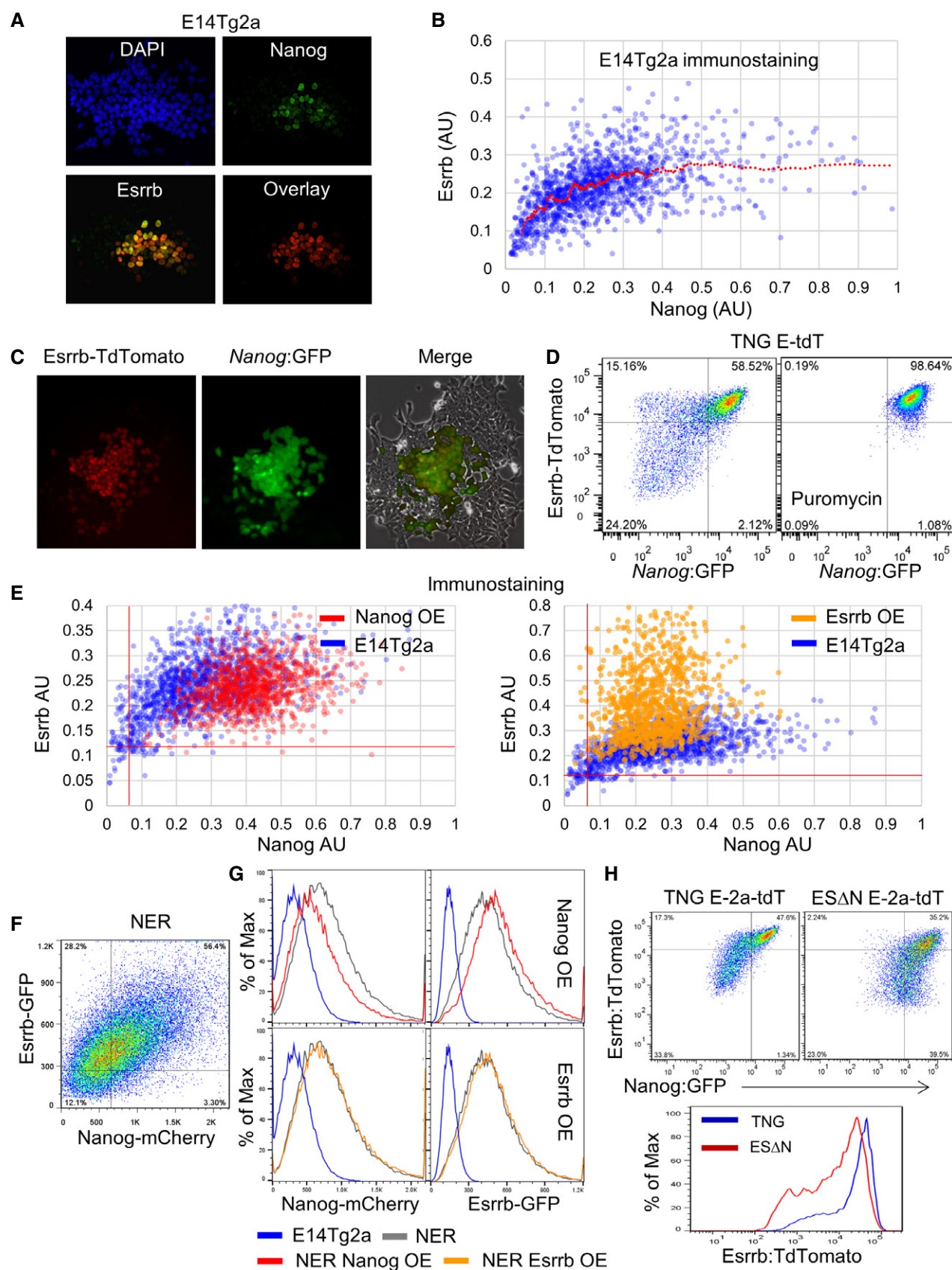

Figure 1.

## The epistatic relationship between NANOG and ESRRB

As shown by relocalization of a NANOG-ERT[2] fusion protein to the nucleus of *Nanog*-null ESCs, *Esrrb* is a NANOG target gene (Festuccia *et al*, 2012). Moreover, quantitation of pre-mRNA shows that *Esrrb* transcription responds to NANOG relocalization within minutes, indicating that *Esrrb* is a primary direct NANOG target. Transient transfection assays also indicate that ESRRB can contribute to luciferase reporter expression driven by the *Nanog* promoter (van den Berg *et al*, 2008). To further investigate the effect of ESRRB on *Nanog* gene expression and of NANOG on *Esrrb* gene expression, immunofluorescence analysis was performed on ESCs overexpressing NANOG or ESRRB. NANOG overexpression resulted in homogeneously robust expression of ESRRB protein in all cells (Fig 1E). This confirms that NANOG is a major positive regulator of *Esrrb*. Upon ESRRB overexpression, the dynamic range of NANOG expression was narrowed, but cells displaying low NANOG protein levels remained detectable (Fig 1E). To explore this reciprocal influence in greater detail, an ESC line in which the endogenous NANOG and ESRRB proteins were fused with GFP and mCherry, respectively, was generated (Appendix Fig S2E and F). Analysis of reporter expression indicated an overall correlation between Nanog and Esrrb levels, but also confirmed the presence of Esrrb-GFP[high]/Nanog-mCherry[low] and the relative absence of Esrrb-GFP[low]/Nanog-mCherry[high] cells (Fig 1F, Appendix Fig S2F). This reporter line was also used to assess the effect of transcription factor overexpression. This confirmed that NANOG overexpression increased Esrrb-GFP expression, and reduced Nanog-mCherry levels, consistent with autorepression (Navarro *et al*, 2012). In contrast, ESRRB overexpression showed minimal effects (Fig 1G). These results establish that the principal epistatic relationship between the two genes places *Nanog* upstream of *Esrrb*.

To directly assess the effect of NANOG protein on *Esrrb* expression in single cells, an Esrrb-2a-tdTomato reporter was introduced into both *Nanog*[−/−] ESΔN and *Nanog*[+/−] TNG ESCs (Chambers *et al*, 2007; Festuccia *et al*, 2012; Fig EV1, Appendix Fig S1). Both lines have GFP knocked-in to one of the *Nanog* alleles, but in ESΔN ESCs, the second *Nanog* allele is also inactivated (Fig EV1, Appendix Fig S2). TNG E-2a-tdT ESCs have a similar FACS profile to TNG E-tdT ESCs. In ESΔN E-2a-tdT ESCs where NANOG protein is absent, the overall levels of Esrrb-2a-tdTomato expression were reduced and the correlation between GFP and tdTomato seen in GFP[+]/tdTomato[+] TNG E-2a-tdT cells was less apparent (Fig 1H). Moreover, as reported by GFP activity, loss of Esrrb expression was not necessarily coupled to downregulation of the activity of the *Nanog* locus (Fig 1H). These results further highlight the important contribution of NANOG protein to *Esrrb* expression.

Our previous study has shown that *Klf4* is also a direct primary target of NANOG (Festuccia *et al*, 2012). Analysis of KLF4 expression by intracellular cytometry in TNG E-tdT cells suggests that ESCs downregulate KLF4 at a similar rate to Esrrb (Fig EV2A). Selection for *Nanog* expression using puromycin abolished heterogeneous expression of both Esrrb-tdTomato and KLF4 (Fig EV2B). While this result is compatible with placement of Nanog upstream of *Klf4* as well as *Esrrb*, we note that additional positive inputs into *Klf4* exist (Niwa *et al*, 2009).

## Loss of *Esrrb* expression reflects commitment out of the naïve state

To investigate the dynamics of the transition between ESCs expressing differing levels of Esrrb and Nanog, TNG E-tdT cells were sorted into two fractions of > 99.5% purity based on expression of Nanog and Esrrb (Fig 2A). Cells were replated in LIF/FCS and the distribution of Esrrb and Nanog reporter expression determined daily (Fig 2A). To restrict the analysis to undifferentiated cells, only SSEA1[+] cells were examined. While Nanog:GFP[high]/Esrrb-tdTomato[high] cells restored the major starting populations, Nanog:GFP[low]/Esrrb-tdTomato[low] cells were unable to do so. In fact, the reduction in SSEA1[+] cell number following replating of Esrrb-tdTomato[low] cells suggests that these had already initiated differentiation. These results suggest that as Nanog and Esrrb levels decline, the self-renewal capacity of the cells diminishes. Dynamic restoration of heterogeneity in transcription factor expression was also assessed upon release of wild-type ESCs from LIF/2i. An initial reduction in ESRRB expression upon removal of GSK inhibition (Martello *et al*, 2012) coincided with the previously observed expansion of the dynamic range of NANOG expression. After this, the levels of both TFs decreased with NANOG-negative cells appearing at day 2 and ESRRB-negative cells later at day 3 (Fig 2B and C). This heterogeneity in loss of naïve pluripotency markers is in line with the heterogeneous loss of a *Rex1* reporter expression following removal of ESCs from culture in 2i, without LIF (Kalkan *et al*, 2017). To exclude potential confounding effects due to release from LIF/2i, these dynamic alterations were independently assessed by sorting double-positive cells from NER cultured in LIF/FCS (Fig 2D). This confirmed that while the cell population progressively reduced the NANOG levels, ESRRB levels were buffered against change for a day following the initial decline at day 1.

To determine the biological significance of the downregulation of transcription factor expression, quantitative colony-forming assays were performed on sorted TNG E-tdT cells. Wholly undifferentiated colonies were produced efficiently from Nanog:GFP[high] Esrrb-tdTomato[high] cells. In contrast, Nanog:GFP[low]/Esrrb-tdTomato[low] cells produced no alkaline phosphatase-positive colonies in either LIF/FCS or LIF/2i (Fig 2E).

Taken together, the above results reveal how loss of Nanog expression may trigger further changes in TF expression that result in reduced ESC self-renewal efficiency (Chambers *et al*, 2007). In cells expressing low NANOG, silencing of Nanog target genes expressed in naïve pluripotent cells, including *Klf4* and *Esrrb*, triggers commitment to exit the naïve state resulting in loss of colony-forming potential in LIF.

## Molecular changes during loss of Esrrb expression

The above results reveal a close connection between loss of Esrrb expression and the progressive extinction of ESC self-renewal. To explore the molecular mechanisms underpinning loss of naïve pluripotency in ESC cultures, a reporter was generated to enable Esrrb downregulation to be monitored with minimal temporal delay by targeting a 2a-destabilized GFP cassette downstream of the ORF at both *Esrrb* alleles (E-GFPd1 cells; Fig EV1, Appendix Fig S1). To generate a heterogeneous population, ESCs were replated at $2 \times 10^3$/cm$^2$ and analyzed after 3 days. FACS analysis of correctly

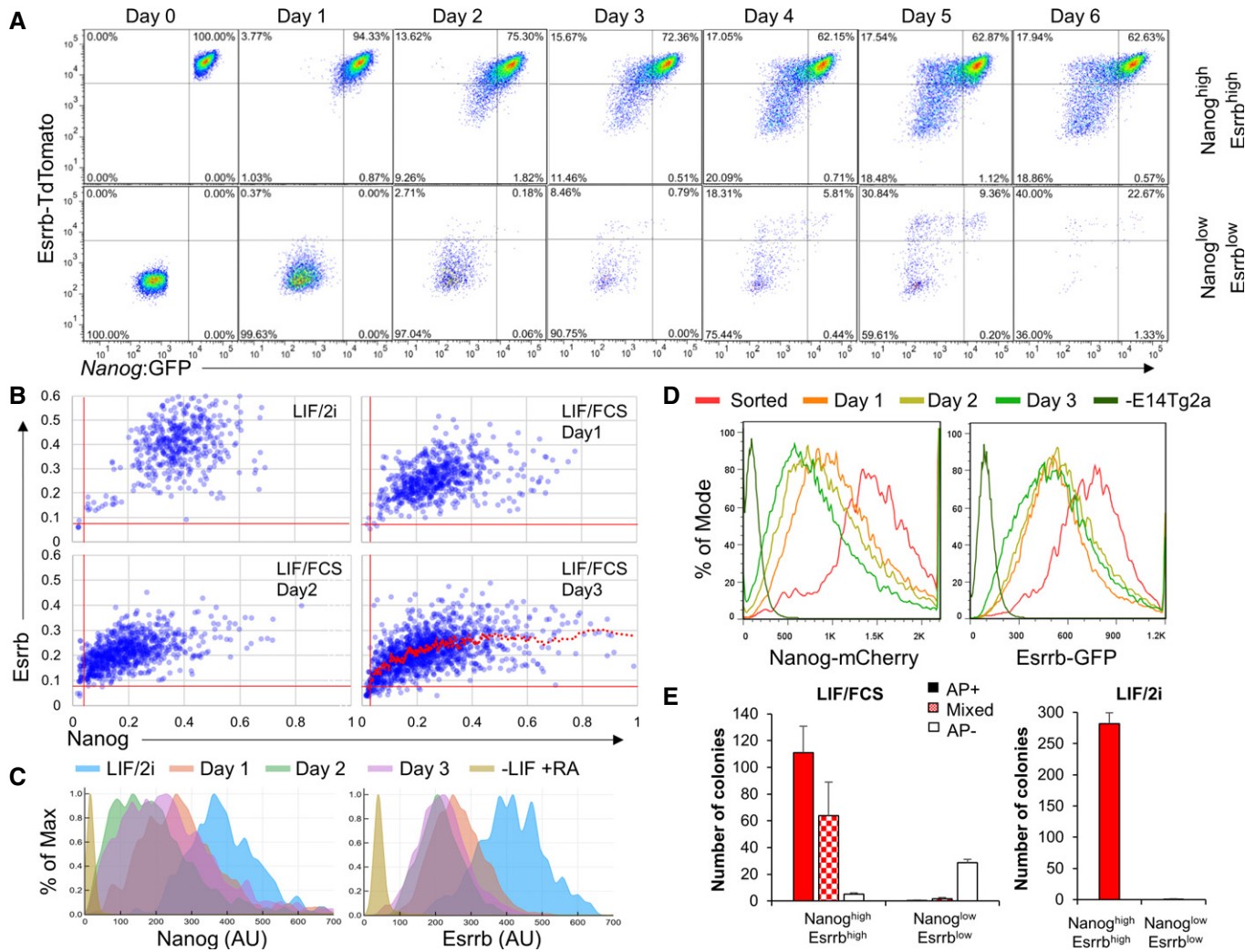

**Figure 2.  Decreased Esrrb expression marks ESC commitment.**

A    SSEA-1⁺ TNG E-tdT ESCs were cultured for 3 days in LIF/FCS and sorted into GFP⁺/tdT^high (Nanog^high/Esrrb^high) or GFP⁻/tdT^low fractions (Nanog^low/Esrrb^low). The purity of the sorted populations (> 99.5%) is shown (day 0). After replating in LIF/FCS, SSEA-1⁺ cells were analyzed daily for Esrrb-tdTomato and Nanog:GFP expression.

B, C    Immunofluorescent detection of NANOG and ESRRB protein in wild-type E14Tg2a ESCs cultured in LIF/2i and released in LIF/FCS for 1, 2, or 3 days. Quantification of the mean Nanog and Esrrb fluorescence levels measured in nuclei identified by automatic segmentation of single optical plane images obtained by confocal microscopy. OCT4^negative cells, also identified by immunostaining, were excluded from the analysis. (B) The red vertical and horizontal lines indicate the negative thresholds defined by staining ESCs differentiated for 3 days in the absence of LIF and the presence of retinoic acid. The red trendline represents the moving average of the data distribution at day 3. Representative of two independent experiments each including at least 1,000 nuclei. (C) Histogram plots of the data presented in (B).

D    Sorted SSEA-1⁺ and Esrrb-GFP^high/Nanog-mCherry^high NER ESCs were replated in LIF/FCS, and SSEA-1⁺ cells were analyzed daily for Esrrb-GFP and Nanog-mCherry expression. Wild-type E14Tg2a set the negative thresholds.

E    SSEA-1⁺ TNG E-tdT ESCs from the sorted populations used above (panel A, day 0) were replated at clonal density in LIF/FCS or LIF/2i and stained for alkaline phosphatase activity after 6 days. Error bars: standard deviation of the number of colonies observed in two independent experiments.

Data information: For a schematic representation of the reporter allele configuration characteristic of each cell line, please refer to Fig EV1.

targeted ESCs showed that culture in LIF/FCS yielded a broad bimodal expression profile that was resolved to a unimodal expression profile in LIF/2i (Fig 3A), matching the results obtained using Esrrb-tdTomato cells. E-GFPd1 cells cultured in LIF/FCS were purified by FACS into Esrrb-GFP^high, Esrrb-GFP^medium, and Esrrb-GFP^negative fractions of SSEA1⁺ cells (Fig 3B). qRT-PCR analysis showed that Esrrb mRNA levels reflected changes in the GFP

reporter levels in the sorted fractions (Fig 3C). Clonal self-renewal assays revealed a distinction in phenotype of Esrrb-GFP^negative cells. While there was no substantial difference between Esrrb-GFP^high and Esrrb-GFP^medium cells in colony-forming capacity in either LIF/FCS or LIF/2i, Esrrb-GFP^negative cells were essentially unable to form fully undifferentiated colonies in either condition (Fig 3D). The behavior of purified E-GFPd1 ESCs following replating in LIF/FCS

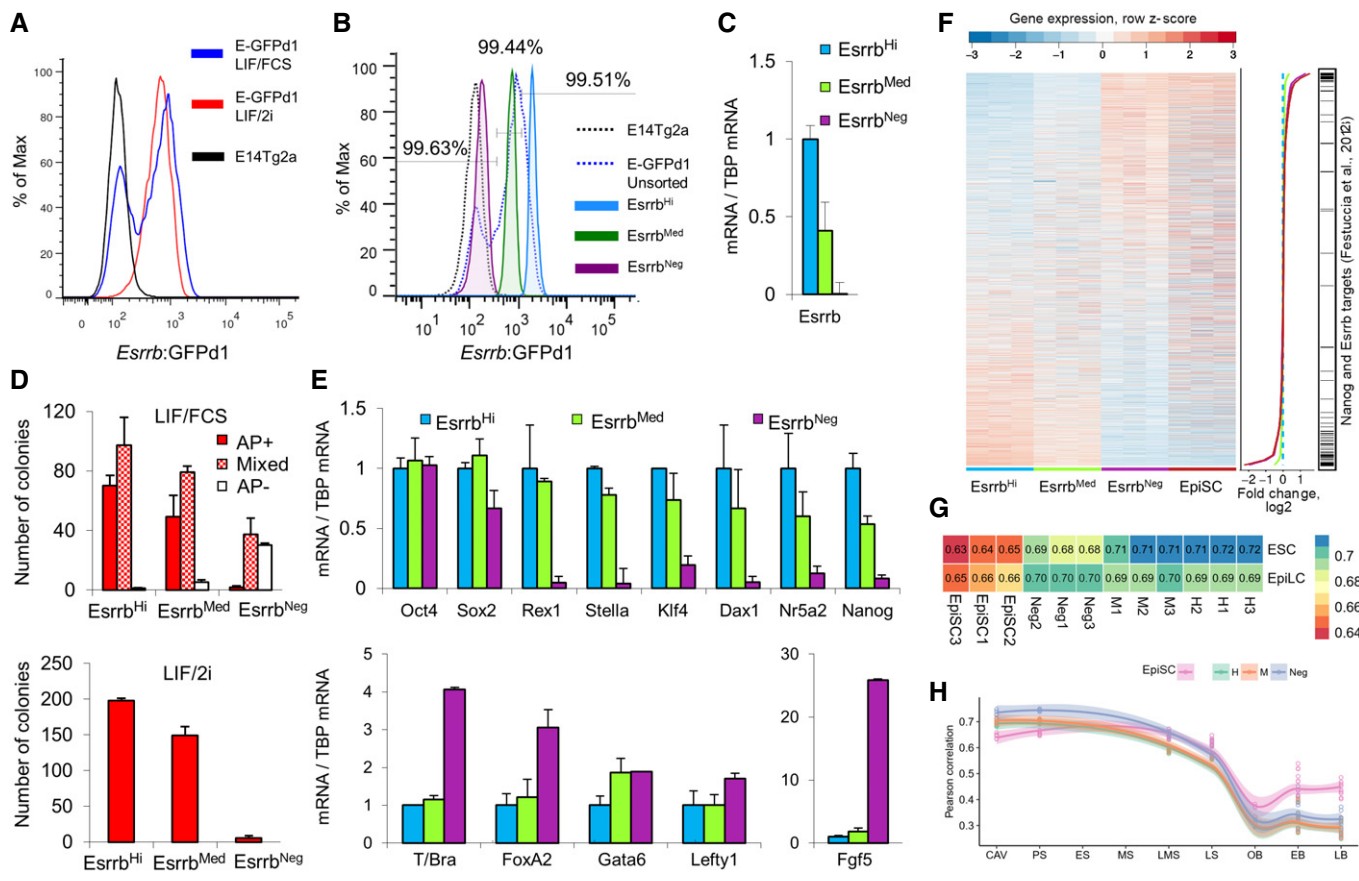

**Figure 3. Naïve pluripotency gene expression is lost upon Esrrb downregulation.**

A  Flow cytometric analysis of GFP expression in SSEA-1[+] E-GFPd1 ESCs cultured in LIF/2i or LIF/FCS compared to wild-type E14Tg2a ESCs.

B  SSEA-1[+] E-GFPd1 ESCs sorted into Esrrb[Hi], Esrrb[Med], or Esrrb[Neg] subpopulations. Purity of the sorted populations, assessed by flow cytometry, is indicated above the sorting gates.

C  Quantitative *Esrrb* mRNA expression analysis of populations sorted in (B). Error bars: standard deviation of gene expression values measured in four independent experiments.

D  Populations sorted in (B) were replated at clonal density in LIF/FCS (top) or LIF/2i (bottom) and scored for alkaline phosphatase after 6 days. Error bars: standard deviation of the number of colonies observed in two independent experiments.

E  Quantitative mRNA expression analysis of populations sorted in (B). Error bars: standard deviation of gene expression values measured in four independent experiments.

F  Microarray analysis of populations sorted in (B) represented as a heatmap ordered by expression change between Esrrb[Hi] and Esrrb[Neg]. Expression levels were measured from three independent experiments and all genes expressed in at least one sample are shown (*n* = 14,553). The central line plot summarizes the average fold change per cell population compared to Esrrb[Hi] cells (log2 scale) with the color coding indicated below the heat map. Previously identified NANOG and ESRRB (Festuccia *et al*, 2012) targets are indicated on the right.

G  Pearson correlation coefficients of the gene expression profiles of Esrrb[Hi] (H), Esrrb[Med] (M), Esrrb[Neg] (Neg) E-GFPd1 ESCs, and embryo-derived (late bud stage) TNG E-2a-tdT EpiSCs (EpiSC) compared to those of naïve ESC and EpiLC from Buecker *et al* (2014).

H  Pearson correlation coefficients in gene expression obtained comparing Esrrb[Hi] (H), Esrrb[Med] (M), Esrrb[−] (Neg) E-GFPd1 ESCs, and embryo-derived (late bud stage) TNG E-2a-tdT EpiSCs (EpiSC) to epiblast/ectoderm cells from embryos dissected at successive developmental stages (CAV, cavity; PS, prestreak; ES, early streak; MS, midstreak; LMS, late midstreak; LS, late streak; OB/EB, no bud/early bud; and LB, late bud). This analysis was restricted to the top 500 differentially expressed gene identified at successive developmental stages sorting by Hotelling T² score, as described in Kojima *et al* (2014).

Data information: For a schematic representation of the reporter allele configuration characteristic of each cell line, please refer to Fig EV1.

culture also supported a distinct phenotype of the Esrrb-GFP[negative] cells, with Esrrb-GFP[high] and Esrrb-GFP[medium] cells rapidly interconverting and generating negative cells, and the Esrrb-GFP[negative] population essentially unable to regain Esrrb expression (Fig EV3A). Coupled with the previous observation that NANOG[low] cells can express distinct ESRRB levels, this indirectly indicates that downregulation of Nanog does not immediately result in the loss of self-renewal ability of ESC. NANOG[low]/ESRRB[positive] cells can

self-renew, with loss of this ability occurring only upon further downregulation of ESRRB.

Fractionated E-GFPd1 ESCs were next analyzed by gene expression profiling to determine the transcriptional changes accompanying phenotypic dismantling of naïve pluripotency. Quantitative mRNA analysis showed that differences in expression between the Esrrb-GFP[high] and Esrrb-GFP[medium] fractions were confined to *Stella* and *Nanog* (Fig 3E). In contrast, Esrrb-GFP[negative] cells showed

complete loss of expression of a range of pluripotency markers typical of the pre-implantation epiblast, but retained *Oct4* and *Sox2* expression, confirming that these cells remained pluripotent. Microarray analyses further indicated that the majority of known NANOG and ESRRB target genes (Festuccia *et al*, 2012) are among the genes with the most dramatic change in expression during the transition from Esrrb-GFP$^{high/medium}$ to Esrrb-GFP$^{negative}$ cells (55.6% of NANOG or ESRRB targets are in the top/bottom 400 genes ranked by the expression difference between Esrrb-GFP$^{high}$ and Esrrb-GFP$^{negative}$ cells; hypergeometric $P = 1.22 \times 10^{-35}$; Fig 3F, Table EV1). Analysis of the regulatory regions in proximity of Differentially Expressed Genes (DEGs) in publicly available ChIP-Seq datasets showed enrichment for binding of NANOG, ESRRB, KLF4, and the mediator subunit MED1 (Chen *et al*, 2008; Fig EV3B), suggesting that these DEGs are stringently controlled by enhancers through which naïve pluripotency TFs act. Taken together, these results suggest that modulation of the entire transcriptional programs controlled by NANOG and ESRRB accompanies commitment to differentiation.

Strikingly, Esrrb-GFP$^{negative}$ cells had also upregulated mRNAs characteristic of primed pluripotent cells (Fig 3E) and globally presented a transcriptional state closer to that of EpiSC (Fig 3F). Moreover, Pearson's correlation coefficient shows that Esrrb-GFP$^{negative}$ cells are transcriptionally closer to primed EpiLC than naïve ESCs ($r$[Esrrb-GFP$^{negative}$, EpiLC] = 0.70, $r$[Esrrb-GFP$^{negative}$, ESCs] = 0.68) while the opposite is the case for Esrrb-GFP$^{high}$ cells (Fig 3G). To further investigate this, sorted cells were replated in N2B27/Activin/FGF and transcripts analyzed after 0, 8, and 24 h. Relative to Esrrb-GFP$^{high}$ cells, Esrrb-GFP$^{negative}$ cells showed faster induction of mRNAs characteristic of primed pluripotent cells (Fig EV3C). This supports the notion that ESCs that have lost Esrrb expression have initiated the transition from naïve to primed pluripotency. Comparison of the gene expression profile of sorted Esrrb-GFP populations and EpiSCs with the transcriptome of epiblast cells from different stages of post-implantation development (Kojima *et al*, 2014) revealed that Esrrb-GFP$^{negative}$ cells are closer transcriptionally to *in vivo* epiblast cells prior to gastrulation. In contrast, EpiSCs more closely resemble epiblast cells from later gastrulating embryos (Fig 3H). To substantiate this observation, we compared the levels of transcripts expressed differentially before (CAV/PS) or after the onset of embryonic gastrulation (LMS, LS) with sorted E-GFPd1 ESCs and EpiSC (Fig EV3D). Genes upregulated after gastrulation have

low expression in both Esrrb-GFP$^{high}$ and Esrrb-GFP$^{negative}$ cells, whereas mRNAs silenced after gastrulation show high expression in both Esrrb-GFP$^{high}$ and Esrrb-GFP$^{negative}$ cells. In contrast, EpiSCs express both mRNA classes at levels intermediate between those detected in pre- and post-gastrulation embryos, suggesting that they have progressed further in development than ESCs fractionated on the basis of Esrrb expression. Esrrb-GFP$^{negative}$ cells therefore capture a pluripotent state characteristic of early post-implantation development.

Gene Ontology (GO) analysis supported the above distinctions (Fig EV3E). Genes involved in morphogenesis, pattern specification, and metabolic processes are upregulated in EpiSCs compared to Esrrb-GFP$^{negative}$ cells. Morphogenesis terms also characterize the Esrrb-GFP$^{medium}$ to Esrrb-GFP$^{negative}$ transition, with changes in genes associated with neuronal development and cell proliferation also apparent. However, the most dramatic changes were in the Esrrb-GFP$^{high}$ to Esrrb-GFP$^{medium}$ transition, which was dominated by downregulation of genes involved in stem cell maintenance/development, negative regulation of differentiation, and upregulation of chromatin structure terms. It is notable that these GO term changes between Esrrb-GFP$^{high}$ and Esrrb-GFP$^{medium}$ characterize states that are largely reversible (Fig EV3A) presaging functional differences in self-renewal that becomes apparent only later at cellular commitment as cell transition from an Esrrb-GFP$^{medium}$ to Esrrb-GFP$^{negative}$ state.

## DNA methylation changes during loss of Esrrb expression

The preceding results suggest that downregulation of Esrrb is linked to progressive changes in the transcriptional identity of ESCs via specific silencing of genes regulated by naïve pluripotency TFs. Locus-specific DNA methylation at pluripotency gene regulatory regions characterizes loss of naïve pluripotency (Ficz *et al*, 2013; Habibi *et al*, 2013; Hackett *et al*, 2013; Leitch *et al*, 2013). Therefore, to test whether CpG methylation drives the loss of pluripotency associated with Esrrb extinction, CpG DNA methylation was analyzed in fractionated SSEA1$^+$ E-GFPd1 ESCs. Digestion with methylation-sensitive restriction nucleases showed that DNA methylation at the *Nanog* and *Esrrb* promoters was indistinguishable between Esrrb-GFP$^{high}$ and Esrrb-GFP$^{medium}$ cells, with a modest increase seen in Esrrb-GFP$^{negative}$ cells (Figs 4A and EV4A). Similarly, modest increases were also observed in Esrrb-GFP$^{negative}$ cells at the *Stella* and *Rex1* promoters, although the latter was already

---

**Figure 4. Different kinetics of loss of TF binding and accumulation of CpG methylation at key regulatory elements of the pluripotency network.**

A  Percentage of methylated CpG dinucleotides profiled across the *Esrrb* and *Nanog* enhancer and the *Nanog* promoter in sorted SSEA-1$^+$/Esrrb$^{Hi}$, Esrrb$^{Med}$, or Esrrb$^{Neg}$ E-GFPd1 ESCs and TNG E-2a-tdT EpiSC. CpG methylation was assessed by measuring protection from digestion of the *HpaII*, *AciI*, *Hin6I*, or *TaqI* restriction sites (indicated by vertical lines in the gene structure maps derived from the mouse reference genome (mm9). Location is expressed relative to the TSS in ESCs (Chambers, 2004; Festuccia *et al*, 2012). Values represent total CpG methylation levels (5mC + 5hmC). Error bars: standard deviation of the measures in four independent experiments.

B  Methylated CpG dinucleotides across the *Esrrb* enhancer were assessed as in (A). SSEA-1$^+$/Esrrb$^{Hi}$ E-GFPd1 ESCs were sorted and placed back in culture overnight before resorting and methylation analysis relative to Esrrb$^{Neg}$ E-GFPd1 ESCs. Error bars: standard deviation of the measures in three independent experiments.

C  The sorted E-GFPd1 ESCs populations used in (A) were assessed by immunoblot for ESRRB and NANOG (left) or OCT4 (right) by fluorescence-based detection (Licor).

D  Quantitative ChIP-PCR analysis of OCT4, NANOG, and ESRRB binding and histone modifications at the *Esrrb* and *Nanog* enhancers and the *Nanog* promoter in the sorted populations used in (A). Error bars: standard deviation of the measures in two (NANOG, ESRRB, H3K27ac) or three (OCT4, H3K4me3, and H3K4me1) independent ChIP experiments, each performed on pooled chromatin from at least three independently sorted samples. The diagrams at the bottom show the approximate position of the regions analyzed for transcription factor binding or histone modifications (in color or black), along with the relative control genomic locations (gray).

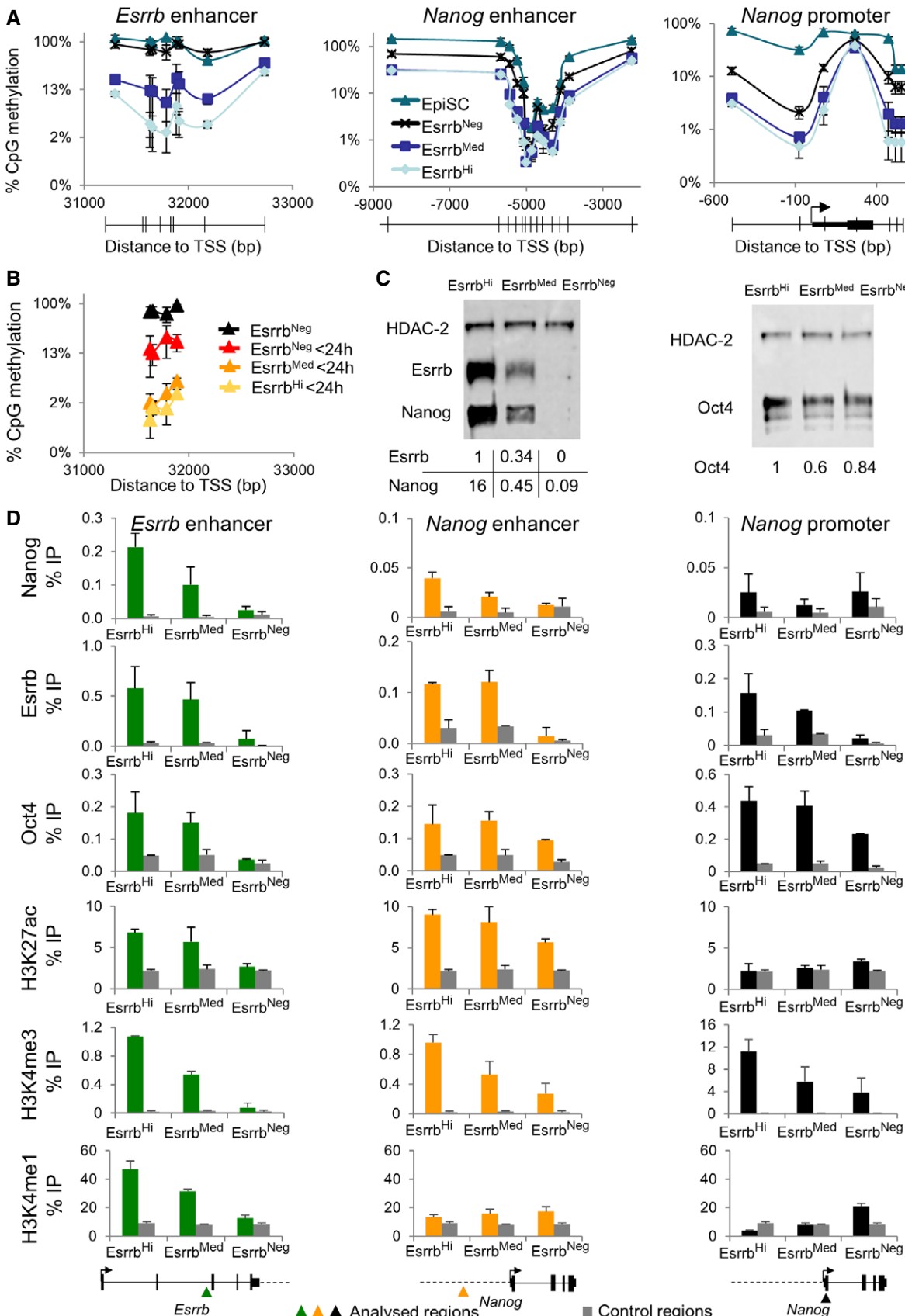

**Figure 4.**

partially methylated in Esrrb-GFP$^{high}$ and Esrrb-GFP$^{medium}$ cells (Fig EV4B and C). These findings agree with the idea that promoters are generally protected from DNA methylation accumulation (Deaton & Bird, 2011), remaining partially unmethylated even when inactive (e.g., *Esrrb* and *Rex1* promoters in EpiSC; Fig EV4A and C). Minimal DNA methylation differences were also seen at the *Nanog* enhancer in ESCs and EpiSCs, where Nanog is expressed (Fig 4A). In contrast, striking changes occur between ESC fractions at the *Esrrb* enhancer, with Esrrb-GFP$^{negative}$ cells and EpiSCs showing almost complete CpG methylation (Figs 4A and EV4A). To investigate DNA methylation dynamics at the *Esrrb* enhancer, FACS-purified Esrrb-GFP$^{high}$ cells were replated in FCS/LIF. Within 24 h, Esrrb-GFP$^{negative}$ cells emerged and FACS purification at this time showed that these cells had increased methylation at the *Esrrb* enhancer from ~2 to ~20% (Fig 4B). These observations indicate that DNA methylation is deposited at differing rates at pluripotency loci with the *Esrrb* enhancer accumulating DNA methylation rapidly. Nevertheless, the fact that not all cells that have downregulated Esrrb at 24 h have methylated the *Esrrb* enhancer indicates that DNA methylation does not drive Esrrb extinction.

### Chromatin changes during transition to Esrrb$^{negative}$ state

Next, TF binding in the three ESC populations was assessed in an attempt to identify potential drivers of the differential accumulation of methylation at the *Nanog* and *Esrrb* enhancers. First, NANOG, OCT4, and ESRRB protein levels were examined in sorted E-GFPd1 cells. Quantitative immunoblot analysis showed that ESRRB protein levels reflected changes in GFP fluorescence and both ESRRB and NANOG protein levels agreed with the corresponding transcript levels in fractionated cells (Figs 3C and E, and 4C). In line with the immunoblot data, NANOG showed a graded ChIP profile at the *Nanog* and *Esrrb* enhancers (Fig 4D). This was also the case for ESRRB binding to the *Nanog* promoter. ESRRB was also lost from both the *Nanog* and *Esrrb* enhancers in Esrrb-GFP$^{negative}$ cells but showed similar enrichment in Esrrb-GFP$^{high}$ and Esrrb-GFP$^{medium}$ fractions, suggestive of a threshold level of ESRRB binding for function at these regulatory elements (Fig 4D). In contrast, OCT4, which is present at similar levels in each fraction, showed distinct patterns of binding in Esrrb-GFP$^{negative}$ cells. Whereas OCT4 binding to the *Esrrb* enhancer was completely abolished in the Esrrb-GFP$^{negative}$ cells, these cells retained detectable binding of OCT4 to the *Nanog* enhancer and promoter (Fig 4D). Coupled to the observation that the Esrrb enhancer accumulated DNA methylation, whereas the Nanog enhancer was protected, these results suggest the hypothesis that the *Nanog* and *Esrrb* enhancers are examples of two classes of regulatory regions that respectively remain active or are decommissioned during early stages of commitment.

Histone modifications were next assessed for their ability to distinguish between these regulatory elements. Active enhancers are marked by H3K27ac, with a fraction also harboring low levels of H3K4me3, a signature of active promoters. H3K4me1 also broadly marks distal regulatory regions. In contrast, H3K27me3 has been proposed to accumulate on silent but poised enhancers and promoters of repressed genes (Heintzman *et al*, 2007, 2009; Rada-Iglesias *et al*, 2011). Therefore, enrichment for H3K4me1, H3K4me3, H3K27ac, and H3K27me3 at the *Esrrb* and *Nanog* enhancers/promoters was determined in the three ESC populations. H3K27me3

was detectable at low levels at the *Nanog* promoter but not at either enhancer (Appendix Fig S3). Both enhancers were instead marked by H3K27ac in Esrrb-GFP$^{high}$ cells: While reduced H3K27ac in Esrrb-GFP$^{negative}$ cells at the *Nanog* enhancer mirrored reduced OCT4 occupancy (Fig 4D), possibly accounting for reduced *Nanog* expression (Figs 3E and 4C and D), the absence of both H3K27ac and OCT4 at the *Esrrb* enhancer in Esrrb-GFP$^{negative}$ cells is consistent with *Esrrb* enhancer decommissioning. H3K4me3 was enriched at the *Nanog* and *Esrrb* promoters, progressively declining at both genes to reach minimal levels in Esrrb-GFP$^{negative}$ cells (Fig 4D, Appendix Fig S3), in line with gene expression (Figs 3C and E, and 4C). H3K4me3 was also detected in Esrrb-GFP$^{high}$ cells at the *Esrrb* and *Nanog* enhancers. This mark was completely lost at the *Esrrb* enhancer in Esrrb-GFP$^{negative}$ cells, suggestive of functional inactivation. Further substantiating the decommissioning of this regulatory element, H3K4me1 was detected at the *Esrrb* enhancer in the Esrrb-GFP$^{high}$ fraction and lost in Esrrb-GFP$^{negative}$ cells.

### Genome-wide analysis of TF binding during transition to Esrrb$^{negative}$ state

To determine the degree to which these findings extended to global changes in enhancer activity occurring during commitment from naïve pluripotency, genome-wide profiles of OCT4 or NANOG binding were determined by ChIP-Seq in sorted fractions of E-GFPd1 cells (Table EV2). Regions bound by OCT4 or NANOG that reduced binding to ≤ 50% of the level in the highest binding population were identified as peaks that lost binding in the population concerned (e.g., in Esrrb-GFP$^{negative}$ cells, OCT4 binding at *Tet2* is 5% of the level in Esrrb-GFP$^{high}$ cells, so the peak is considered lost, Fig 5H). In Esrrb-GFP$^{high}$ cells, OCT4 and NANOG each bound to ~10,000 regions (Fig 5A and C). In Esrrb-GFP$^{negative}$ cells, the number of OCT4 binding sites was only reduced slightly (Fig 5A), whereas the number of NANOG-bound regions was reduced drastically (Fig 5C), consistent with a reduced NANOG protein level (Fig 4C). Analysis of the average profiles of OCT4 occupancy revealed that regulatory elements either lost or maintained OCT4 binding during the transition to Esrrb-GFP$^{negative}$: the positions for which an OCT4 peak was called only in negative cells ($n = 3,532$) already showed detectable OCT4 binding in Esrrb-GFP$^{high}$ cells (Fig 5B). In contrast, NANOG displayed a marked reduction in binding, globally, even at the 286 positions that were still identified as peaks in Esrrb-GFP$^{negative}$ cells (Fig 5D). Likewise, the number of sites co-bound by OCT4 and NANOG was also dramatically lower in Esrrb-GFP$^{negative}$ cells relative to Esrrb-GFP$^{high}$ cells (Fig 5E).

To compare the functional consequences of loss or maintenance of binding of NANOG and OCT4 at a comparable set of regulatory elements, the chromatin regions bound by both factors in Esrrb-GFP$^{high}$ cells were further analyzed. Of the 3,675 regions bound by both OCT4 and NANOG in Esrrb-GFP$^{high}$ cells, two major classes of peaks were identified. Both classes of regulatory elements lost NANOG in Esrrb-GFP$^{negative}$ but could be distinguished by whether OCT4 binding was lost (Class I elements, $n = 2,034$; exemplified by a *Tcfcp2l1* enhancer) or not (Class II elements, $n = 1,565$; exemplified by one of the *Sox2* enhancers; Fig 5F). Importantly, differentially expressed genes in the proximity (distance to closest gene ≤ 20 kb; Table EV3) of Class I elements showed decreased expression in Esrrb-GFP$^{negative}$ ESCs or EpiSC ($n_{\leq 20\ kb} = 947$;

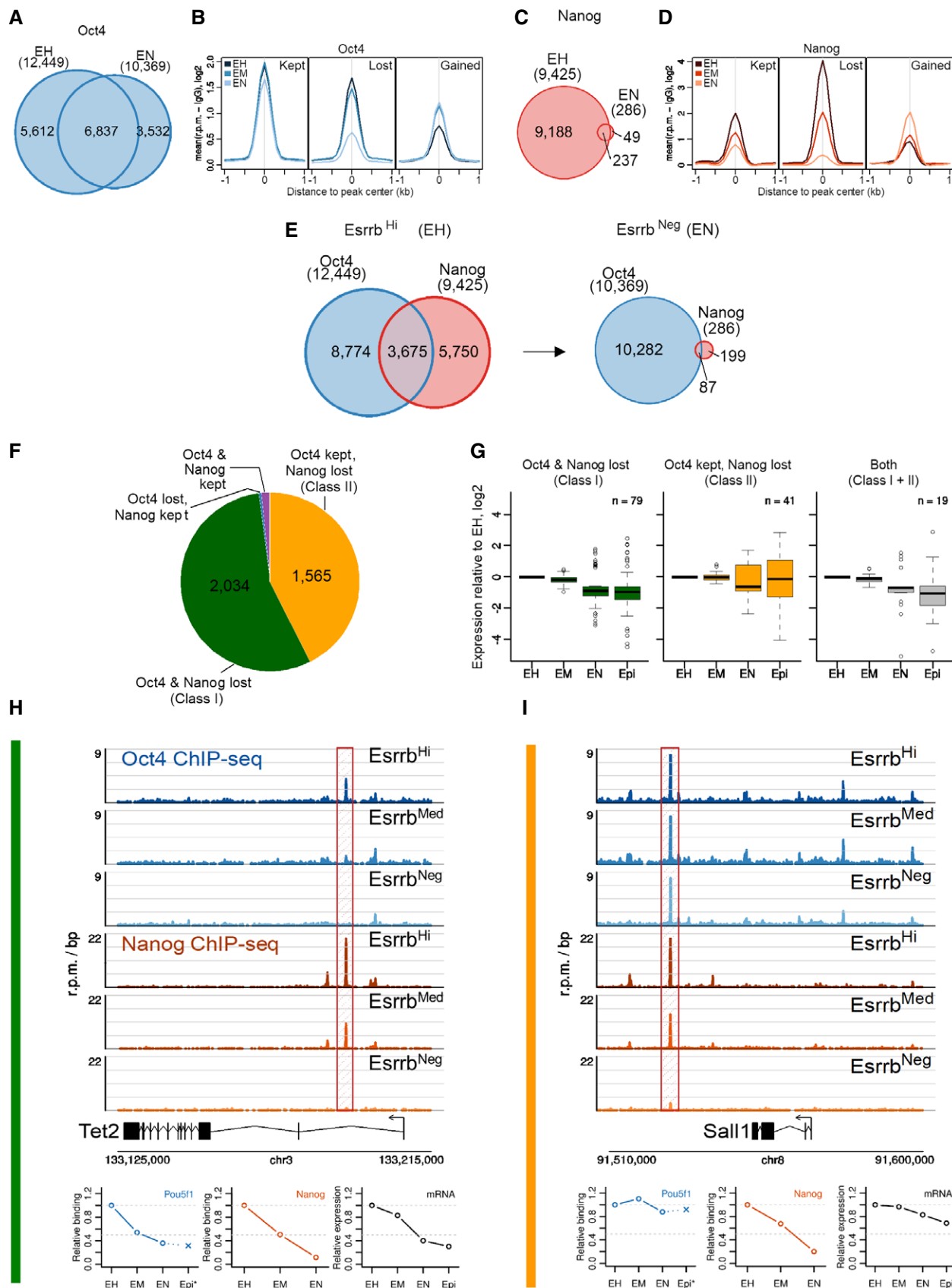

Figure 5.

**Figure 5.  Global profiles of OCT4 and NANOG chromatin binding during Esrrb downregulation.**

A–D  (A, C) Venn diagram showing overlap of (A) OCT4- or (C) NANOG-bound regions (see main text for peak calling strategy) in sorted Esrrb^Hi (EH) and Esrrb^Neg (EN) E-GFPd1 ESCs. (B, D) Average binding profile of (B) OCT4 or (D) NANOG in binding peaks where binding in Esrrb^Neg ESCs was kept, lost, or gained relative to Esrrb^Hi E-GFPd1 ESCs. Read counts per base pair were normalized by library size and by subtracting the background (IgG) signal. Esrrb^Hi (EH), Esrrb^Med (EM), and Esrrb^Neg (EN).

E  Venn diagram showing the overlap of NANOG- and OCT4-bound regions in sorted Esrrb^Hi and Esrrb^Neg E-GFPd1 ESCs.

F  Pie chart showing the number of genomic regions co-bound by NANOG and OCT4 in Esrrb^Hi E-GFPd1 ESCs that lose binding of NANOG and OCT4 (Class I) or that lose binding of NANOG but retain OCT4 (Class II) during transition to Esrrb^Neg. A few regions keep NANOG binding but lose OCT4 ($n = 16$) or keep binding of both ($n = 60$).

G  Boxplots showing the fold change in expression levels of genes proximal to Class I elements, Class II elements, or both Class I and Class II elements in Esrrb^Hi (EH), Esrrb^Med (EM), or Esrrb^Neg (EN) E-GFPd1 ESCs and EpiSC (Epi). Boxes span the inter-quartile range (IQR) from the first to the third quartile. The line indicates the median. Whiskers extend up to 1.5 IQR, and outliers are plotted.

H, I  Binding profiles of OCT4 and NANOG in the proximity of two genes representative of (H) Class I (*Tet2*) or (I) Class II (*Sall1*) in Esrrb^Hi, Esrrb^Med, or Esrrb^Neg E-GFPd1 ESCs. The relative occupancies of NANOG and OCT4 at *Tet2* or *Sall1* (positions highlighted by hatched boxes) alongside expression levels of *Tet2* and *Sall1* mRNAs in Esrrb^Hi (EH), Esrrb^Med (EM), or Esrrb^Neg (EN) GFPd1 ESCs and EpiSC (Epi) are shown at the bottom. OCT4-binding data from EpiSC are from Buecker *et al* (2014).

$n_{DEG} = 79$—Fig 5G). In contrast, differentially expressed genes near Class II elements showed a higher propensity to maintain or increase expression ($n_{\leq 20\ kb} = 683$; $n_{DEG} = 41$—Fig 5G). The 19 differentially expressed genes associated with both Class I and Class II elements showed a decreased expression pattern suggesting a dominant role of Class I over Class II elements in controlling gene expression (Fig 5G). Among genes linked to sites losing binding of NANOG and OCT4 (Class I), important regulators of the naïve pluripotent state were identified, for instance, Tet2, which is silenced during the Esrrb-GFP^high to Esrrb-GFP^negative transition (Fig 5H). Other pluripotency factors, such as Sall1, which are expressed in both Esrrb-GFP^high and Esrrb-GFP^negative ESCs or EpiSC, maintained OCT4 binding in all populations, including EpiLC (Buecker *et al*, 2014; Fig 5I).

Comparison with publicly available TF-binding datasets in ESCs (Chen *et al*, 2008) revealed that Class I elements are bound by ESRRB, KLF4, and NANOG more robustly than Class II elements (Fig 6A). In contrast, Class I elements bind OCT4 less than Class II elements (Fig 6A). To assess whether Class II elements are more likely to remain active in primed pluripotent cells, available datasets were analyzed to compare the FAIRE signal, H3K27ac, EP300, and OCT4 enrichment in ESC and EpiLC (Buecker *et al*, 2014). Indeed, in primed EpiLCs, Class II elements retain OCT4 binding, a more open chromatin structure and histone mark signatures of active enhancers (Fig 6B). Interestingly, in EpiLCs, Class II elements also show higher enrichment for OTX2 (Fig 6B). This may either reflect a specific role for OTX2 in stabilizing OCT4 binding in primed pluripotent cells or be a consequence of the maintained chromatin accessibility in EpiLC. Overall, these results suggest that retention of OCT4 binding identifies regulatory elements that remain active following early stage differentiation. Consistent with this, differential motif discovery analysis performed using RSAT (Thomas-Chollier *et al*, 2011) on DNA sequences underlying Class I and Class II peaks identified an octamer-binding motif at Class II peaks (Fig 6C). Notably, the Oct-Sox motif was not identified in this differential analysis suggesting that it is equally present in both peak classes. In contrast, consensus-binding motifs for ESRRB and KLF factors were enriched at Class I peaks (Fig 6C). These observations suggest that binding of OCT4 to Class I elements is labile and dependent on co-binding by naïve pluripotency TFs. This may explain their early downregulation during the process of commitment.

## Discussion

### TF heterogeneity and the transition between naïve and primed pluripotency

NANOG is the pluripotency TF for which a direct quantitative correlation between levels of expression and ESC self-renewal efficiency has been most clearly demonstrated (Chambers *et al*, 2003, 2007; Abranches *et al*, 2014). Accordingly, NANOG is considered to act as a rheostat to tune ESC self-renewal efficiency (Mullin & Chambers, 2012). Modulation of NANOG activity in ESCs alters transcription of a limited number of direct target genes (Festuccia *et al*, 2012). Among these, *Esrrb* is the most strongly upregulated TF. Consistent with this, ESRRB is able to mediate several downstream effects of NANOG, both in pluripotent cells and primordial germ cells (Festuccia *et al*, 2012; Zhang *et al*, 2018). NANOG protein is expressed in the nuclei of ESCs cultured in LIF/FCS in a heterogeneous manner, a feature shared with several other pluripotency proteins, including ESRRB (Chambers *et al*, 2007; Torres-Padilla & Chambers, 2014).

To clarify the role of ESRRB on pluripotency gene regulatory network function, subpopulations of ESCs expressing distinct ESRRB levels were analyzed for ESC self-renewal, target gene expression, DNA methylation, and chromatin binding by TFs. In contrast to reduced NANOG expression, which is a reversible state (Chambers *et al*, 2007; Abranches *et al*, 2014; Filipczyk *et al*, 2015), loss of ESRRB commits cells to exit the naïve state. This can be clearly seen in clonal self-renewal assays where Esrrb^negative cells are essentially unable to form colonies in LIF. We therefore examined how NANOG and ESRRB downregulation are coordinated in single ESC to trigger

**Figure 6.  Differential TF binding and activity of Class I and Class II elements in ESC and EpiLC.**

A  Average binding profile of ESRRB, KLF4, NANOG, OCT4, and SOX2 to Class I (green), Class II (orange), or all NANOG- and OCT4-bound (gray dashed line) elements in ESCs [data from Chen *et al* (2008)]. Read counts per base pair were normalized by library size and by subtracting the background (IgG) signal.

B  Average FAIRE profile or ChIP-Seq profiles for H3K27ac, OCT4, Ep300, or OTX2 in ESCs or EpiLC for Class I and Class II elements [data from Buecker *et al* (2014)].

C  Results of a differential motif analysis (Thomas-Chollier *et al*, 2011) in Class I and Class II peaks.

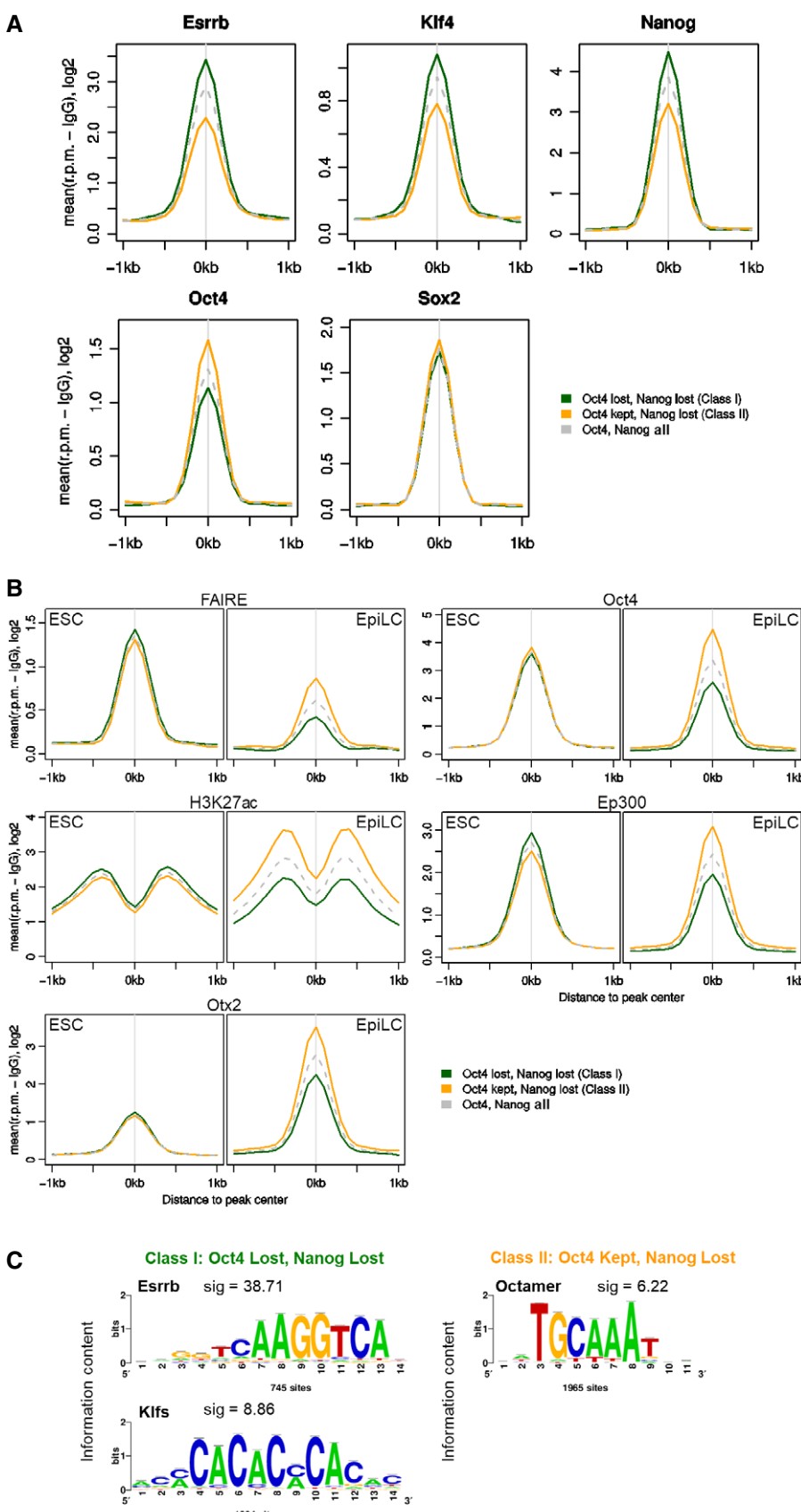

**Figure 6.**

differentiation. Compared to ESRRB, NANOG has a higher dynamic expression range, with ESCs accommodating significant downregulation of NANOG without drastic changes in ESRRB: only when NANOG levels drop below a certain threshold is *Esrrb* extinction possible. Therefore, the NANOG[low] compartment is composed of two subpopulations, which differ in ESRRB expression and self-renewal ability. Cell-tracking experiments have also identified two distinct subpopulations of ESCs that have downregulated NANOG but differ in cell fate and differentiation potential (Filipczyk *et al*, 2015; Hastreiter *et al*, 2018). Our results extend these findings by showing that ESRRB can distinguish these subpopulations. This suggests that when unsupported by the transcriptional activity of NANOG, the naïve pluripotency gene regulatory network is weakened. While alternative positive inputs into *Esrrb* expression have been documented (Martello *et al*, 2012), our results indicate that such inputs ultimately fail in the absence of NANOG. ESCs may then initiate downregulation of other naïve pluripotency TFs including *Klf4* and *Esrrb*. Indeed, modulation of the entire NANOG-dependent transcriptional program may enable the ordered dismantling of naïve pluripotency. In this respect, heterogeneity in NANOG might be qualitatively distinct from heterogeneity in ESRRB and other pluripotency TFs.

A coordinated downregulation of naïve pluripotency markers is reminiscent of the process occurring during the transition from pre- to post-implantation development (Boroviak *et al*, 2014, 2015; Kojima *et al*, 2014). The period of loss of NANOG expression occurring at peri-implantation (Chambers *et al*, 2003; Acampora *et al*, 2013) may enable cells of the epiblast to downregulate a number of pivotal naïve pluripotency determinants, including ESRRB (Adachi *et al*, 2013). In the post-implantation epiblast, NANOG is re-expressed (Hart *et al*, 2004; Osorno *et al*, 2012; Hoffman *et al*, 2013) but ESRRB is not (Adachi *et al*, 2013), likely due to differences in signaling environments between pre- and post-implantation epiblasts. It is noteworthy that subpopulations of ESC cultures have been proposed to bear a similar character to primed pluripotent cells and vice versa (Hayashi *et al*, 2008; Han *et al*, 2010). Upon differentiation, naïve ESCs transit through a state (named EpiLCs) with transcriptional similarity to the early post-implantation epiblast (Buecker *et al*, 2010; Hayashi *et al*, 2011). Esrrb-GFP[negative] cells and EpiLCs have a high correlation in OCT4 chromatin binding (Pearson's $r = 0.60$). Both Esrrb-GFP[negative] cells and EpiLCs also have a high correlation in gene expression to pre- or early-streak epiblasts. In contrast, EpiSCs have a higher correlation to later gastrulation stages. Similar correlations with embryonic expression programs have been reported in studies investigating the molecular changes accompanying heterogeneous downregulation of *Rex1* during naïve ESC differentiation (Kalkan *et al*, 2017). Therefore, TF heterogeneity represents a useful model with which to study the transition between the regulatory configurations sustaining pre-and post-implantation pluripotency.

## Using Esrrb expression as a tool to study dynamic chromatin reorganization

We applied our differentiation model to characterize how changing TF activity re-shapes chromatin at two crucial regulatory elements showing divergent behavior during the conversion from naive to primed pluripotency, initially by assessing DNA methylation.

The *Nanog* enhancer (Blinka *et al*, 2016) which is active in both naïve ESCs and EpiSC is unmethylated in both. In contrast, the *Esrrb* enhancer (Festuccia *et al*, 2012; Moorthy *et al*, 2017), which is active in ESCs but inactive in primed pluripotent cells, becomes completely methylated as cells commit to differentiate. Our results build on previous reports that accumulation of DNA methylation at the regulatory regions of pluripotency genes accompanies loss of naïve pluripotency (Ficz *et al*, 2013; Habibi *et al*, 2013; Hackett *et al*, 2013; Leitch *et al*, 2013). Moreover, our results illustrate the progressive nature by which accumulation of DNA methylation occurs, as exemplified by the intermediate levels of methylation detected at the *Esrrb* enhancer in Esrrb-GFP[medium] ESC. Furthermore, the slower kinetics of 5mCpG accumulation at the *Esrrb* enhancer relative to loss of ESRRB expression favors the view that loss of TF binding precedes DNA methylation and that increased DNA methylation is a consequence of, rather than a cause of, loss of TF binding. This is in line with the suggestion the DNA methylation is not a major driver of naïve pluripotency loss (Kalkan *et al*, 2017). Moreover, the primacy of TF binding relative to changes in DNA methylation has also been proposed during the Esrrb-induced reprogramming of EpiSCs to a naïve state (Adachi *et al*, 2018). The ability of ESC self-renewal to withstand complete loss of the DNA methylation machinery, with cell lethality of DNMT triple-knockout ESCs only becoming apparent during differentiation (Tsumura *et al*, 2006), suggests that the increase in DNA methylation levels in committed cells reflects a change in the mode of gene regulation, from a dynamic process chiefly directed by TFs in pluripotent cells to one stabilized by DNA methylation (Festuccia *et al*, 2017).

How TF binding relates to deposition of specific histone modifications was also assessed. In line with loss of TF binding and the associated ability to recruit co-activators, H3K27ac was lost from the enhancer of *Esrrb* but not *Nanog*. Similarly to acetylation, H3K4me3, for which a correlation with enhancer activity is less well established (Pekowska *et al*, 2011; Outchkourov *et al*, 2013; Shen *et al*, 2016; Henriques *et al*, 2018), was also maintained in Esrrb-GFP[negative] cells at the enhancer of *Nanog* but not *Esrrb*. Loss of H3K4me1 has been proposed to be a key step in enhancer decommissioning (preprint: Agarwal *et al*, 2017; Cao *et al*, 2018; Whyte *et al*, 2012; Yan *et al*, 2018), though a direct function for this mark remains controversial (Dorighi *et al*, 2017; Cao *et al*, 2018; Yan *et al*, 2018). Since inactivation of various MLL family members fails to ablate H3K4 methylation at most pluripotency enhancers or alter pluripotency gene expression (Denissov *et al*, 2014; Wang *et al*, 2016; Cao *et al*, 2017, 2018; Dorighi *et al*, 2017; Yan *et al*, 2018), the importance of this class of histone modification in regulating the activity of the pluripotency network remains to be stringently tested. Our finding that H3K4me1 becomes undetectable at the *Esrrb* enhancer in committed cells, coincident with loss of TF binding and gain of DNA methylation, might provide a well-characterized model to address these controversies.

## Loss of TF binding drives the ordered dismantling of naïve pluripotency

The process of cell type specification occurring during development has been described as a reshaping of the activity of the enhancer pool available in different conditions. While some enhancers are

conserved from pluripotent to differentiated cell types, others are lost or created (Stergachis *et al*, 2013). Indeed, a noteworthy conclusion from our ChIP-Seq analysis is that the early steps in differentiation are dominated by a loss of regulatory elements. This suggests that keeping a vast pool of active or poised regulatory regions might be a defining feature of the pluripotent genome. OCT4 is present at similar levels in all the populations we analyzed. Only a few sites showed increased OCT4 occupancy ($n = 3{,}532$) in Esrrb-GFP[negative] cells, and these had detectable OCT4 binding in Esrrb-GFP[high] cells (Fig 5B). In apparent contradiction to this, activation of new enhancers has been reported to occur during the dismantling of naïve pluripotency (Buecker *et al*, 2014; Factor *et al*, 2014; Yang *et al*, 2014; Cao *et al*, 2018; Yan *et al*, 2018). However, such regulatory elements were not identified here, possibly because they are regulated exclusively by lineage-specific TFs missing from our analysis. Alternatively, the stages of early commitment we capture may be primarily characterized by a loss of activity of pluripotency-specific regulatory regions, with the activation of novel enhancers occurring later in differentiation after transcriptional de-repression of lineage-specific regulators allows their accumulation in the cell. Indeed, FOXD3 has been shown to bind and open new chromatin sites to which OCT4 and additional factors are then recruited (Krishnakumar *et al*, 2016). Similarly, accumulation of OTX2 during exit from naïve pluripotency partially redirects OCT4 to a new

enhancer set (Buecker *et al*, 2014; Yang *et al*, 2014). Since *Otx2* is a direct target gene repressed by NANOG (Festuccia *et al*, 2012), a strong increase in OTX2 levels would occur only after complete loss of NANOG (Acampora *et al*, 2013, 2017). The moderate increase in binding by OCT4 in Esrrb-GFP[negative] cells that we detect might indeed be a sign of such an accumulation and occur at sites where the effect of lineage-specific TFs is not yet fully evident, as compared to other studies. In line with this interpretation, regions with increased OCT4 occupancy during the Esrrb-GFP[high] to Esrrb-GFP[negative] transition also show marginal but detectable OTX2 binding in ESCs and become robustly bound by OTX2 only after 48 h of EpiLC induction (Fig EV5A). In fact, OTX2 is enriched at regions that maintain OCT4 binding in EpiLC (Figs 6B and EV5A), suggesting that at these sites OTX2 and OCT4 reciprocally stabilize binding to DNA (Buecker *et al*, 2014).

We identified two distinct classes of enhancers bound by OCT4 and NANOG in Esrrb-GFP[High] cells: Class I sites lose binding of both NANOG and OCT4 in Esrrb-GFP[negative] cells, while Class II sites lose NANOG but retain OCT4 binding. The selective loss of Class I sites reflects a reconfiguration of protein interactions at crucial regulatory regions and may be the earliest example of the wholesale "pruning" of regulatory elements present in pluripotent cells responsible for the initial canalizing steps away from naïve pluripotency. In this model (Fig 7), declining levels of ESRRB, NANOG, and other TFs

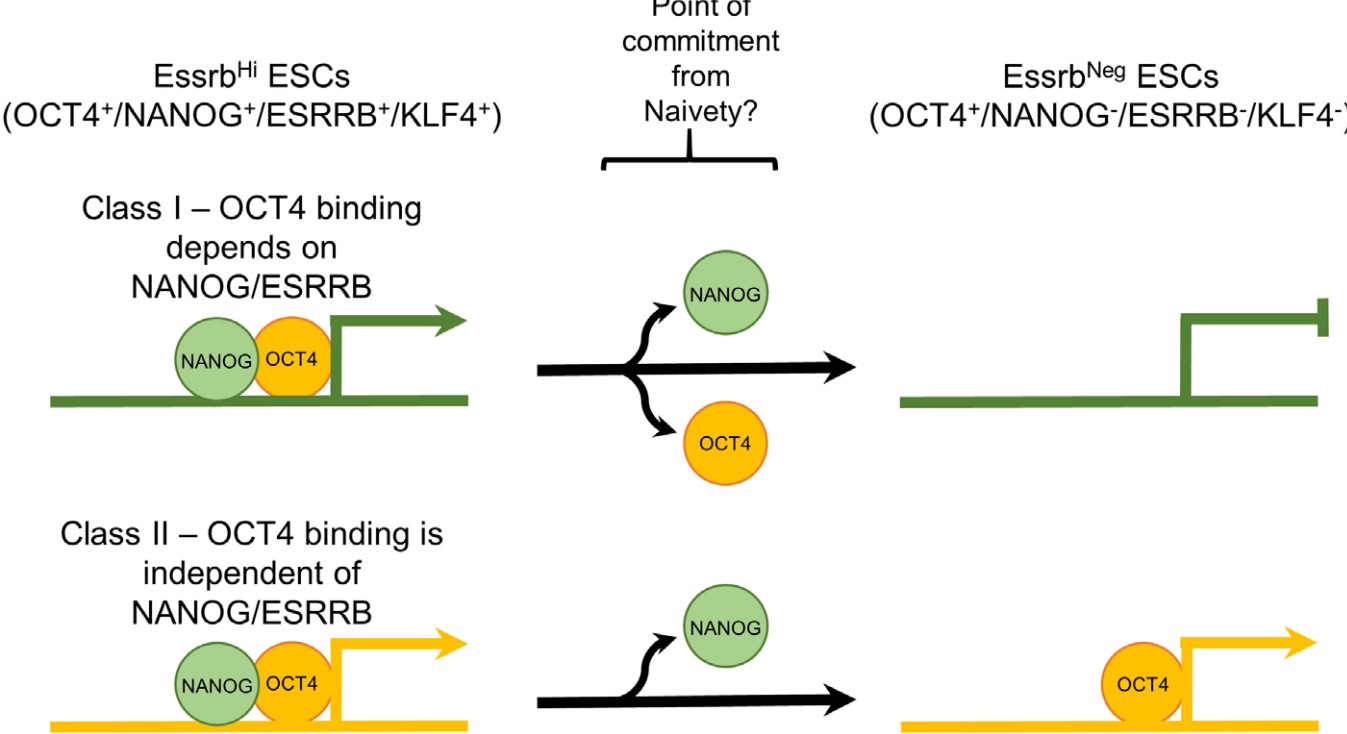

**Figure 7. Model of TF binding to enhancer classes during commitment from naïve pluripotency.**

Diagram summarizing the changes occurring at decommissioned or maintained enhancers during commitment from naïve pluripotency. Decreased NANOG levels, although reversible, facilitate subsequent decrease in the levels of other naïve pluripotency TFs, including Esrrb and Klf4. Downregulation of these TFs triggers irreversible changes in the pool of active enhancers and marks commitment to differentiation. Class I enhancers are regions where OCT4 occupancy depends on the robust binding of naïve TF. These elements are decommissioned during commitment, losing histone acetylation and accumulating DNA methylation. Other enhancers, on average showing less pronounced binding of naïve TFs, retain OCT4 and active histone marks in primed pluripotent cells. These elements constitute a core of "primed" regulatory elements bound by OCT4 that may serve to nucleate the regulatory architecture through which lineage-specific TFs collaborate with Oct4 to specify the early post-implantation epiblast identity.

would lead to loss of activity of regulatory regions in which continued OCT4 binding depends upon binding of naïve pluripotency factors. Such dependency, suggested by the enrichment of naïve TF-binding motifs, is in line with the notion of cooperative DNA binding and transcriptional regulation by TFs (Spitz & Furlong, 2012; Morgunova & Taipale, 2017). However, no individual chromatin modification or naïve pluripotency TF-binding pattern can predictably distinguish regulatory elements that lose or maintain activity during early differentiation. In the future, it will be important to determine the details by which the chromatin organization at Class I and II elements determines differential responses to a common change in the cellular environment. Whether regulatory elements in a class all lose TF binding simultaneously and whether specific regulatory elements lose individual TFs simultaneously or sequentially are also important points for future analysis. Nevertheless, it is likely that early loss of activity at key Class I regulatory elements reflects the crossing of the Rubicon of commitment, triggering further, irreversible, gene regulatory network changes.

The recent report that ESRRB acts as a mitotic bookmark of key regulatory elements during ESC cell division (Festuccia et al, 2016) is relevant to the above model. As mitosis is accompanied by eviction of several TFs from the chromatin, bookmarking by ESRRB may allow the rapid reassembly of the pluripotency network in early G1 (Festuccia et al, 2016). Declining ESRRB levels may therefore impair the reactivation of key pluripotency enhancers after mitosis. This would imply that cell division coincides with the time of commitment for differentiating ESCs, with the first steps of enhancer decommissioning beginning in early G1 (Mummery et al, 1987; Pauklin & Vallier, 2013).

Other mechanisms could potentially contribute to enhancer decommissioning in differentiating ESCs. Recently, it has been reported that FOXD3, a TF expressed in naïve ESCs but upregulated during the transition to EpiSC-like cells, co-occupies numerous enhancers that are also bound by pluripotency TFs. However, FOXD3 directly contributes to the inactivation of only a fraction of these elements (Respuela et al, 2016). A similar number of Class I and Class II enhancers are bound by FoxD3 in ESCs (~30%; Fig EV5B) with both Class I and Class II enhancers binding FOXD3 at similar levels (Fig EV5C). As only Class I enhancers are decommissioned upon ESRRB and NANOG loss, FOXD3 binding is not a major driver of the differential responses identified in this study.

Taken together, the results presented here provide a possible mechanistic explanation for the connection observed between NANOG levels and ESC self-renewal efficiency (Chambers et al, 2007). In LIF/FCS cultures, the NANOG^low population is not uniform but can be fractionated into cells expressing distinct ESRRB levels. ESRRB^positive cells self-renew efficiently, whereas NANOG^low cells expressing the lowest ESRRB levels have lost this ability. Declining concentrations of NANOG and ESRRB may result in a stepwise reduction in the activity of the naive pluripotency gene regulatory network, reflected in a decrease in the number of chromatin sites bound by OCT4. This in turn decreases the pool of active enhancers until the point of commitment is reached, and ESCs are ushered out of the naïve state. The regulatory regions where OCT4 binding depends on naïve TF binding, identified here as Class I enhancers, become decommissioned and accumulate DNA methylation with fast kinetics. Other

enhancers retain OCT4 and active histone marks in primed pluripotent cells. Thus, decreased NANOG levels, although reversible, facilitate subsequent decreases in the levels of other naïve pluripotency TFs, including ESRRB and KLF4, that may then extinguish the entire naïve transcriptional program while leaving a core of "primed" regulatory elements bound by OCT4. This would to ensure that commitment to differentiation occurs in an orderly way that retains the functional activity of the program encoding early post-implantation epiblast identity.

# Materials and Methods

### ESC culture

Embryonic stem cells were cultured in GMEMβ-mercaptoethanol/LIF/10%FCS as described (Smith, 1991) or in N2B27 supplemented where indicated with PD0325901 (1 µM) and CHIRON99021 (3 µM) (Ying et al, 2008). When required, ESCs were differentiated in N2B27/Activin (10 ng/ml)/bFGF (12 ng/ml) or in GMEMβ-mercaptoethanol/10%FCS supplemented with $10^{-6}$ M of all-trans retinoic acid.

### Derivation of E-GFPd1, E-tdT, TNG E-tdT, TNG E-T2a-tdT, and ΔN E-T2a-tdT reporter ESC lines

E-GFPd1 and E-tdT were derived from wild-type E14Tg2a ESCs; TNG E-tdT and TNG E-T2a-tdT were derived from TNG cells (Chambers et al, 2007). ΔN E-T2a-tdT were derived from ESΔN-NERT ESCs (Festuccia et al, 2012). Targeting vectors are constructed as follows: Around 5 kb of homology to the genomic sequence upstream of the end of the coding sequences for *Esrrb* [ESC main isoform—(Festuccia et al, 2012)] is linked by a T2a peptide (for E-GFPd1, TNG E-T2a-tdT, and ΔN E-T2a-tdT ESCs) or a mutated non-cleaving T2a peptide (for TNG E-tdT ESCs) to the coding sequence for GFP or TdTomato followed by—IRES-Hygromycin^R or—IRES-Blasticidin^R cassettes. Resistance cassettes are followed by around 5kb of homology to the genomic sequence downstream of the end of the coding sequence for *Esrrb* (see Fig EV1 and Appendix Fig S1). $10^{7}$ parental line ESCs were resuspended in 700 µl of PBS, mixed with 25 µg of linearized targeting vectors ethanol precipitated and resuspended in 100 µl PBS, and transfected using an electroporator (ElectroSquarePorator, BTX Harvard Apparatus—protocol T820). Cells were plated in ten 10-cm dishes in GMEMβ/LIF/FCS. Forty-eight hours after electroporation, relevant selection was added to the medium (5 µg/ml Blasticidin or 200 µg/ml Hygromycin). The resulting cell colonies were picked and expanded as clonal lines before genomic DNA extraction (as described in the Appendix Supplementary Methods). For homozygous reporter lines, positively targeted clones underwent a second electroporation and selection round using the same targeting vector carrying a distinct resistance cassette.

### Derivation of NER reporter lines

Esrrb-GFP and Nanog-mCherry targeting vectors are constructed as follows: 100/50 bp homologous to the end of the coding sequences for *Nanog* and *Esrrb* are linked by a five glycine linker to a cassette

encoding GFP-T2a-Puromycin[R] and mCherry-T2a-Blasticidin[R], respectively, and followed by 100/50 bp homologous to the beginning of the endogenous 3′ UTR of the gene. Point mutations are introduced in the UTR sequence to ensure that the vectors are immune to cutting by Cas9. The backbone of the vectors also includes a U6 promoter driving expression of a gRNA matching the targeted genomic location. One hour before transfection, $1 \times 10^6$ E14Tg2a ESCs were replated in a well of a six-well plate. Thirty minutes before transfection, 3 μg of Esrrb-GFP targeting vector and 1 μg of plasmid driving expression of Cas9 (Addgene n. 44719) were added to 250 μl of GMEMβ (without FCS), and mixed with 250 μl of GMEMβ (without FCS) containing 3 μl of Lipofectamine 2000 (Invitrogen—cat. 11668-019), pre-incubated for 5 min. The mixture was occasionally mixed by flicking during the 30-min incubation and then added dropwise to the cells plated before. After 1 day, the cells were trypsinized and replated into 10-cm dishes in GMEMβ/LIF/FCS. The following day, puromycin (1 μg/ml) was added to the medium. The resulting cell colonies were picked and expanded as clonal lines before genomic DNA extraction (as described in the Appendix Supplementary Methods). Correct recombination was verified by PCR amplification of the genomic region surrounding the stop codon of *Esrrb*, with primers annealing outside of the homology arms, and sequencing. The selected line is heterozygous for the Esrrb-GFP reporter allele and carries a short deletion (27 bp) in the 3′UTR of the wild-type *Esrrb* allele. Esrrb-GFP ESCs were transfected as before with Nanog-mCherry targeting vectors and correct recombination verified by PCR and sequencing as outlined for *Esrrb*. The resulting NER lines are heterozygous for the Nanog-mCherry reporter allele (see Fig EV1).

### Flow cytometry

All cell lines were plated (2,000/cm$^2$) and cultured without selection for 3 days. For TNG E-tdT ESCs, cells were trypsinized and resuspended at $2 \times 10^6$ cells/ml in 10%FCS/PBS containing anti SSEA-1 mouse monoclonal antibody from ascitic fluids (DSHB Cat # MC-480) diluted 1:1,000. Cells were incubated (on ice, 15 min), washed in ice-cold PBS, resuspended, and incubated in 10%FCS/PBS containing goat anti-mouse IgM-Alexa647 secondary antibody (Molecular Probes Cat # A-21238; on ice, 15 min), after washing in ice-cold PBS, cells were resuspended in 10%FCS/PBS and analyzed using a LSR II flow cytometer system (Becton, Dickinson). For NER ESCs, cells were trypsinized, resuspended at $1 \times 10^6$ cells/ml in GMEMβ/LIF/FCS containing anti SSEA-1 mouse monoclonal antibody conjugated to Alexa405 (BD X) diluted 1:1,000, and incubated on ice for 15 min. Cells were spun down, resuspended in GMEMβ/LIF/FCS, and analyzed using a LSR II flow cytometer system.

### Sorting and replating of fluorescent reporter lines

Embryonic stem cells stained for SSEA-1 as described above were purified using a FACSAria cell sorter (Becton, Dickinson) and post-sort cell purity determined using a LSR II Fortessa flow cytometer (Becton, Dickinson). For TNG E-tdT ESCs timecourse experiments, cells were replated in GMEMβ/LIF/FCS in separate wells of 24-well plates and cultured for the indicated time. Cells were harvested every day, stained for SSEA-1, and analyzed on a LSR II Fortessa flow cytometer. Sorted Nanog:GFP$^{high}$/Esrrb-tdTomato$^{high}$ or

Nanog:GFP$^{low}$/Esrrb-tdTomato$^{low}$ cells were trypsinized and replated at days 1, 2, and 3 for analysis at days 4, 5, and 6, respectively, to prevent overgrowth. For NER and E-GFPd1 ESCs timecourse experiments, Esrrb-GFP$^{high}$/Nanog-mCherry$^{high}$ (NER) or Esrrb-GFPd1$^{high}$, Esrrb-GFPd1$^{medium}$ and Esrrb-GFPd1$^{negative}$ (E-GFPd1) cells were replated in GMEMβ/LIF/FCS in separate wells of six-well plates and cultured for the indicated time. Cells were harvested every day, stained for SSEA-1, and analyzed on a LSR II Fortessa flow cytometer. All data were analyzed using the FlowJo software suite (Tree Star). For all clonal density plating experiments in GMEMβ/LIF/FCS, 600 sorted cells were replated in duplicate in gelatinized six-well plates and cultured in the indicated conditions for 7 days prior to colony scoring. For plating in N2B27/LIF/2i, cells were replated as above in six-well plates that had been coated overnight with poly-L-ornithine 0.01% (Sigma Cat # P4957), washed, and coated 2 h with 5 μg/ml laminin (Millipore, Cat # CC095) in PBS.

### Chromatin preparation

$10^7$ to $5 \times 10^7$ E14Tg2a Esrrb-2a-GFPd-IBIH ESCs were resuspended at $3.3 \times 10^6$ cells/ml in pre-warmed GMEMβ/LIF/FCS and cross-linked for 10 min at RT with 1% formaldehyde (Sigma Cat # F8775-25ML) in the dark. Cross-linking was stopped by adding 0.125 mM glycine (5 min, RT), cells were pelleted (5 min, 100 rcf, 4°C), washed twice with ice-cold PBS, and resuspended in 10%FCS/PBS at $10^7$ cells/ml. Keeping the samples refrigerated, Esrrb-GFP$^{high}$ (Top 10%), Esrrb-GFP$^{medium}$ (15% of the distribution immediately above negative), and Esrrb-GFP$^{negative}$ cells were purified using a FACSAria cell sorter (Becton, Dickinson). Since gradual loss of fluorescence was observed in fixed cells after prolonged incubation, multiple batches of freshly fixed cells were sorted for a maximum of 45 min, a time when loss of fluorescence was not yet noticeable. After sorting, cell purity was determined using the same instrument. Cells were spun down, transferred to 1.5 ml Eppendorf tubes, washed with cold PBS, counted, and resuspended at $5 \times 10^6$ cells/ml in swelling buffer (5 mM Pipes pH 8, 85 mM KCl) freshly supplemented with 1× protease inhibitor cocktail (Roche Cat # 04 693 116 001)/0.5% NP-40. After 30 min on ice with occasional shaking, nuclei were centrifuged (600 *g*, 5 min, 4°C), washed in TSE (0.1% SDS, 1% Triton, 2 mM EDTA, 20 mM Tris-HCl pH 8)/150 mM NaCl, freshly supplemented with 1× protease inhibitor cocktail, and resuspended in the same buffer at $10^7$ cells/ml. Samples were sonicated in 1.5 ml tubes using a Bioruptor (Diagenode; 4 × 10 min cycles (each divided into 30 s ON-30 s OFF subcycles) at maximum power in circulating ice-cold water. Chromatin was then centrifuged (30 min, 20,000 rcf, 4°C) and the supernatant stored (−80°C) until use. 5 μl was used to quantify chromatin concentration and check the DNA fragment size on a 1.5% w/v 0.5× TAE agarose gel. The typical fragment size was 100–250 bp.

### ChIP-Seq data analysis

ChIP-Seq data were uploaded to GeneProf (Halbritter *et al*, 2012) and analyzed using the typical workflows. After alignment, the SISSRs (Jothi *et al*, 2008) were used to call enrichment-binding events ("peaks"). To assess variability in ChIP-Seq peak occupancy

between cell states, the amount of ChIP-Seq binding signal was calculated for each dataset in all peaks called for at least one dataset (after merging overlapping peaks). A peak was called as "lost" only if less than 50% of its maximum observed binding was retained (normalizing differences in sequencing library depth). To identify DNA motifs differentially enriched in different classes of peaks, we used the "oligos_6–8nt" algorithm of the RSAT web server (February 2017 version) using either Class I peaks as foreground and Class II peaks as background or vice versa (van Helden *et al*, 1998; Thomas-Chollier *et al*, 2011, 2012).

Microarray data were uploaded to the same GeneProf experiment, and array probes were mapped to the internal gene database. Data from several other studies were also imported, all of which were available from the GeneProf database (Halbritter *et al*, 2014). These data were used primarily to examine the binding profiles of other transcription factors and DNA-associated proteins in the proximity of ChIP-Seq peaks identified in this study. An overview is given in the Appendix Table S1.

### Statistical analysis

No explicit power calculations or distribution tests were performed, but we designed our experiments according to community standards and using established tools and protocols. Results are presented as mean values + or ± standard deviations. In all instances where results are presented as a mean value, the number of biological replicates performed for each experiment is stated in the figure legend. Statistical tests were used for the identification of differentially expressed genes in the microarray data using limma (Ritchie *et al*, 2015), for the identification of binding peaks from ChIP-Seq data using SISSRs (Jothi *et al*, 2008), and for the *de novo* DNA motif analysis based on ChIP-Seq peaks using RSAT (van Helden *et al*, 1998; Thomas-Chollier *et al*, 2011, 2012). Please refer to the respective sections of the Materials and Methods for details.

## Data availability

High-throughput data from this study have been submitted to the Gene Expression Omnibus (GEO) under accession number GSE118907 (https://www.ncbi.nlm.nih.gov/geo/query/acc.cgi?acc = GSE118907).

**Expanded View** for this article is available online.

### Acknowledgements

We thank Kirsten Liggat and the staff of the MRC CRM flow cytometry facility for providing technical assistance. We thank Pablo Navarro (Institut Pasteur) for supporting NF to perform a part of the revision experiments. Research in IC's laboratory is supported by the Medical Research Council of the UK, by the Human Frontier Sciences Program, and by an MRC centenary award to NF.

### Author contributions

NF and IC conceived the study and designed experiments. NF performed experiments and analyzed the data. FH performed bioinformatics analyses and with SRT assisted in data interpretation. AC performed RNA analyses, AG performed immunoblot analyses, and DC provided technical assistance. NF and IC wrote the manuscript.

### Conflict of interest

The authors declare that they have no conflict of interest.

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
