## [Review Process File · The EMBO Journal]

Esrrb extinction triggers dismantling of naïve pluripotency and marks commitment to differentiation

Nicola Festuccia, Florian Halbritter, Andrea Corsinotti, Alessia Gagliardi, Douglas Colby, Simon R. Tomlinson and Ian Chambers

Review timeline:

Submission date:	14th Dec 2016
Editorial Decision:	22nd Feb 2017
Revision received:	26th Jun 2018
Editorial Decision:	13th Jul 2018
Revision received:	24th Aug 2018
Accepted:	27th Aug 2018

Editor: Daniel Klimmeck

Transaction Report:

1st Editorial Decision

22nd Feb 2017

Thank you for the submission of your manuscript (EMBOJ-2016-95476) to The EMBO Journal. My apologies for the extended duration of the review process of the manuscript at this time of the year. As outlined earlier, two referees have originally been assigned to your manuscript, however one of them did not come back to us even after repeated messages, so we had to request input from a third referee, which delayed the review, but was needed to comprehensively assess your work. We have now received reports from two referees, which I copy below.

As you will see, both reviewers highlight the potential interest and novelty of your work, although they also express a number of major concerns that will have to be addressed before they can support publication of your manuscript in The EMBO Journal. In particular, referee #1 points out that some conclusions are not sufficiently well supported by the current data (ref #1, pts. 5,6), in particular referring to the hierarchy between Nanog and Esrrb, which in his/her view needs to be investigated in greater depth (ref#1, pts 8, 13). This referee also states the need for you to provide further proof of the physiological relevance of your findings for the in vivo development (ref#1, first paragraph). Further, referee #2 agrees in that some conclusions are not convincing at this point and consolidating work is needed including on the mutual causality between Esrrb and Nanog (ref#2, pt. 1,2). In addition, the referees list a number of technical issues and controls that would need to be conclusively addressed to achieve the level of robustness needed for The EMBO Journal.

From my side, I judge the referee comments to be generally reasonable, thus we are in principle happy to invite you to submit a revised manuscript addressing the referees' comments.

Please note that while the point on the in vivo relevance of your in vitro findings is in principle well taken, I also realise that the mentioned debate is a broader theme, whose clarification might be well beyond the scope of the current study. Nevertheless I would appreciate your feedback on this particular point.

REFeree COMMENTS:

Referee #1:

In this manuscript, the authors report the characterization of the subpopulations of mouse embryonic stem cells (mESCs) identified by the expressions of Nanog and Esrrb. They found that the fluorescent reporters monitoring Nanog and Esrrb allowed separation of Nanog⁺/Esrrb⁺, Nanog⁻/Esrrb⁺ and Nanog⁻/Esrrb⁻ subpopulations. Among them, Nanog⁻/Esrrb⁻ showed minimal capacity for self-renewal, suggesting their character prone to extinction from pluripotent state. Analyses of Esrrb-high, -medium and -low fractions for gene expression, DNA methylation of particular regulatory elements and TF binding to the genome demonstrated that Esrrb-low cells turned off the naïve-specific genes with partial acquisition of DNA methylation signatures. Two classes of Oct4 binding sites were revealed that showed differential character in Esrrb-high and -low subpopulations, in which the Class I elements lost the binding in the Esrrb-low cells whereas the class II elements keep the occupation. The comparison to the previous data set suggested that the character of the Esrrb-low cells is close to that of the primed state of pluripotent stem cells, indicating that it could be regarded as the transition state from naïve to primed pluripotency. The reversible transition between the different states in mESCs has been reported in several markers. This is an interesting phenomenon *in vitro* and possibly correlates to the *in vivo* development at peri-implantation period. However, direct evidence for a link between the mESC culture system and the *in vivo* developmental process is still missing. This study provides several interesting findings but still lacks the confirmation of the connection to the *in vivo* status. Even *in vitro*, it is still controversial how to mimic the transition from naïve to primed state. EpiSCs and EpiLCs were reported as the primed state model but it is impossible to make direct conversion from naïve to primed state in culture. I hope the publication of this report after revision will provide an important piece of the knowledge to understand the molecular mechanism that manages the transition and will contribute to the establishing the ideal transition system in future for further analysis.

1. Is the pattern of the merges between Nanog and Esrrb reporters consistent to immuno-staining data?
2. Page 3; 'Naïve pluripotency is a characteristic of pre-implantation epiblast cells and their *in vitro* derivatives, embryonic stem cells (ESCs).'*---*This is true for mouse but not for human.
3. Page 3; Heterogeneous expression of Tbx3 was first reported in Niwa et al, 2009.
4. Page 3; The authors stated that EpiSCs are apparently equivalent of the post-implantation epiblast, but it was reported that the character of mouse EpiSCs is most similar to the anterior primitive streak that is more specified than neutral epiblast state (Kojima et al, Cell Stem Cell, 2014).
5. In the comparison between the naïve and primed state of pluripotency, how do the authors explain about the fact of the Nanog expression in both states? Is it reactivated in the primed state?
6. Page 6; The authors proposed that Nanog positions at upstream of Esrrb based on the over-expression data. How about their link in knockout ES cells? How could this idea be consistent with the presence of Nanog-negative; Esrrb-positive cells?
7. Page 8; The authors stated that they analyzed the SSEA-1+ cells. Did SSEA-1+ selection cover all Nanog-positive or Esrrb-positive subpopulations? Is any bias ruled out here?
8. Page 9; The authors proposed that Nanog acts upstream of Esrrb and Klf4 since the selection for Nanog expression resulted in their homogeneous expression. However, this could also be explained by parallel position or Nanog positioning downstream of Esrrb and Klf4.
9. Page 11; The authors emphasized the close relation between Esrrb-negative cells and EpiSCs. Is it possible to make EpiSC-like cells from Esrrb-negative cells easier than the unselected ES cells *in vitro*?

10. Page 12; 'upregulated during the Esrrb-GFP negative to EpiSC transition'---this is confusing statement due to the lack of evidence for direct transition.

11. In relation to Figure 5, how about the binding of Oct4 and Nanog in EpiSCs in their relationships to the patterns observed in Esrrb-high and low cells?

12. Page 17; The authors described that the motif analysis with RSAT resulted in over-representation of ESRRB, KLF4 and KLF5 binding sites at Class I peaks and Oct4 binding site at Class II peaks. I am not familiar with RSAT analysis, and I wonder how these results can be interpreted. Do they the only represent sequences over-represented to oct4 binding sites of each class? Why not Oct-Sox motifs? Are there any motifs over-represented nearby (~100 bp) the Class II peaks?

13. For the regulation of Nanog and Esrrb by extrinsic signals, it was reported that Nanog expression is sensitive to the MAPK signal whereas Esrrb is preferentially activated by Gsk3 inhibition, suggesting their differential regulation. Although the authors argued about the hierarchy between them, it will be possible that they are just in parallel or non-hierarchical relation.

Referee #2:

Festuccia et al report on the differences in Esrrb expression states in mouse ES cells, and their relationship with loss of naïve pluripotency. The authors document that Esrrb defines three different populations within ESC cultures under serum/LIF conditions, based on the expression levels of Esrrb. The negative Esrrb population is proposed as the population which first exits naïve pluripotency. This conclusion is supported by a number of experiments based on reporter, and endogenous expression of Nanog and Esrrb, as well as by colony formation and AP analysis. In addition, the authors show that the enhancers of Esrrb and Nanog show indeed gradual changes in the Esrrb-high, low, and negative, suggesting that the Esrrb-low are indeed of intermediate nature upon pluripotency exit.

The data presented is carefully controlled and the manuscript for the most part well written according to the data presented. The manuscript is novel, and presents new insights in the understanding of transitional states upon exit from pluripotency. It identifies Esrrb expression as a marker of the earliest step upon exit and therefore as the 'first gatekeeper' of naïve pluripotency. These are all new data that will have a big impact on the stem cell community. I am therefore fully supportive for publication in EMBO J, provided the authors address my following points.

1. On page 7, related to Figure 2. Because Nanog-low and Esrrb-low cells are within the same population, it is impossible to disentangle who is doing what (e.g. which population 'precedes' which population). Towards this end, can the authors perform the same analysis as in Figure 2A (dayly analysis of fluorescence) of Nanog expression in Esrrb-/- cells and viceversa (probably an inducible shRNA or siRNA approach may be needed)? The data presented in the supplementary Figure S4 provides a partial answer to this issue, but based on this data, I disagree with the authors in their conclusions that 'loss of Esrrb expression was uncoupled from down regulation of the activity of the Nanog locus' (Page9).

2. Related to Figure 5, on page 16: the authors conclude that "Nanog displayed a marked reduction in binding to its targets globally in Esrrb-neg cells". This is unsurprising, considering that NANOG protein is practically gone in these cells (as shown in the WB on Figure 4C). It is also unsurprising that ChIP for Nanog and Esrrb shows a reduction in their binding in the ChIP presented in Figure 4D in the Esrrb-negative cells, so the extent of the relevance of a reduction in binding here is questionable, since it is basically the expression that goes down, not the binding (e.g. the binding is not regulated, their expression is). In contrast, the analysis of H3K27ac is much stronger and rather interesting. Can the authors complement this ChIP with additional enhancer and promoter marks (e.g. H3K4me1, H3K122ac and/or H3K4me3)?

3. Same applies for the genes that loose Nanog binding in Figure 5D and G. How many peaks 'gained' Nanog binding in Figure 5D, are these numbers really relevant?

Minor comments.

1. Some sentences are a little overstated, so please rephrase (example, page 4, '...enabled transcriptional and epigenetic changes accompanying extinction of the naïve pluripotency network to be identified'): the authors only looked at changes in DNA methylation by restriction enzyme/PCR, and H3K27ac was only looked at on the Nanog and Esrrb genes), no global 'epigenetic' analysis is performed thoroughly. Another example is on page 9, first two lines of the third paragraph).
2. The nomenclature of H3K27Ac and H3K27(me)3 is incorrect (it should be H3K27ac and H3K27me3).
3. Throughout the figures. It would be beneficial to add small 'diagrams' of the reporters used in the panels, as well as the genotype of the cell lines in the figures (e.g. Nanog +/-, Esrrb - wt, etc.): the nomenclature EF4, E14Tg2a, TNG ...etc is well known by only few people. Considering the broad readership of EMBO J., the authors could make an effort to make the data more accessible to a wider community.
4. Figure S3 is not numbered in order in the manuscript.
5. The identification of the DNA binding motifs in Figure 6C is not meaningful unless p-values are presented.
6. Data presented in Figure S4 is quite important, it would be best to include it in the main manuscript.

1st Revision - authors' response

26th Jun 2018

Thanks you for your patience with this revision. We have addressed the vast majority of the points raised by the reviewers. In particular we have: quantified immunofluorescence staining and added data from new reporter cell lines (including cells with overexpression of either NANOG or ESRRB); compared the kinetics of acquisition of peri-implantation epiblast transcripts in ESRRB^{negative} and ESRRB^{high} cells; extended our analysis of histone modifications. Thanks to implementation of our reviewers' suggestions we think the manuscript is much improved and I hope you may now consider it suitable for publication.

As regards the in vivo situation with Esrrb and Nanog expression, we know that expression of both Nanog and Esrrb are lost at the peri-implantation period (there is extensive published data on this point for NANOG and we know that ESRRB mRNA is not present in the peri-implantation embryo). In addition, enforced expression of either prevents effective differentiation. During post-implantation development, Nanog but not Esrrb is re-expressed. To provide additional in vivo context for our work we compared the expression profiles of ESRRB^{negative} and ESRRB^{high} cells to the profiles of embryo populations post-implantation. These observations suggest that the down-regulation of NANOG and ESRRB that we are studying is of physiological importance.

Referee #1:

In this manuscript, the authors report the characterization of the subpopulations of mouse embryonic stem cells (mESCs) identified by the expressions of Nanog and Esrrb. They found that the fluorescent reporters monitoring Nanog and Esrrb allowed separation of Nanog+/Esrrb+, Nnaog-/Esrrb+ and Nanog-/Esrrb- subpopulations. Among them, Nanog-/Esrrb- showed minimal capacity for self-renewal, suggesting their character prone to extinction from pluripotent state. Analyses of Esrrb-high, -medium and -low fractions for gene expression, DNA methylation of particular regulatory elements and TF binding to the genome demonstrated that Esrrb-low cells turned off the naïve-specific genes with partial acquisition of DNA methylation signatures. Two classes of Oct4 binding sites were revealed that showed differential character in Esrrb-high and -low subpopulations, in which the Class I elements lost the binding in the Esrrb-low cells whereas the class II elements keep the occupation. The comparison to the previous data set suggested that the character of the Esrrb-low cells is close to that of the primed state of pluripotent stem cells,

indicating that it could be regarded as the transition state from naïve to primed pluripotency. The reversible transition between the different states in mESCs has been reported in several markers. This is an interesting phenomenon in vitro and possibly correlates to the in vivo development at peri-implantation period. However, direct evidence for a link between the mESC culture system and the in vivo developmental process is still missing. This study provides several interesting findings but still lacks the confirmation of the connection to the in vivo status. Even in vitro, it is still controversial how to mimic the transition from naïve to primed state. EpiSCs and EpiLCs were reported as the primed state model but it is impossible to make direct conversion from naïve to primed state in culture. I hope the publication of this report after revision will provide an important piece of the knowledge to understand the molecular mechanism that manages the transition and will contribute to the establishing the ideal transition system in future for further analysis.

We are pleased that this reviewer supports publication of our manuscript and recognises that our work can provide knowledge to understand the transition out of naïve pluripotency.

1. Is the pattern of the merges between Nanog and Esrrb reporters consistent to immuno-staining data?

A comparison of Esrrb-TdTomato with ESRRB immunofluorescence is presented in Fig S3C. We have previously presented a dual NANOG immunofluorescence of TNG cells (Chambers et al. 2007). To further investigate whether Esrrb is really downregulated after Nanog, we have quantitatively assessed immunofluorescence for ESRRB and NANOG (Fig 1B). This provides further evidence for the existence of ESRRB-high, NANOG-low cells but not ESRRB-low, NANOG-high cells.

We have also made a new reporter line based on Nanog-mCherry and Esrrb-gfp fusion proteins (NER cells). The new data (Fig. 1F, G and Fig. S3E, F) show that there are cells in the population that have reduced Nanog more strongly than Esrrb but the new data also suggests that Nanog-low; Esrrb-high cells are less prominent than previous data suggested. Therefore, although these new data do not alter the main point of our study, we have tempered our claims throughout the manuscript.

Importantly, the new data support the absence of a significant population of Esrrb-low; Nanog-high cells.

2. Page 3; 'Naïve pluripotency is a characteristic of pre-implantation epiblast cells and their in vitro derivatives, embryonic stem cells (ESCs).'-This is true for mouse but not for human.

We have changed the text on page 3 to read 'Naïve pluripotency is a characteristic of pre-implantation mouse epiblast cells and their in vitro derivatives, embryonic stem cells (ESCs).'

3. Page 3; Heterogeneous expression of Tbx3 was first reported in Niwa et al, 2009.

We thank the reviewer for pointing out this error. Although we had cited the Niwa et al paper earlier in the sentence we have now amended the text to properly and more explicitly credit Niwa et al with the original Tbx3 discovery.

4. Page 3; The authors stated that EpiSCs are apparently equivalent of the post-implantation epiblast, but it was reported that the character of mouse EpiSCs is most similar to the anterior primitive streak that is more specified than neutral epiblast state (Kojima et al, Cell Stem Cell, 2014).

We now include a comparison between our transcriptional profiling and the data from Kojima et al. In the last paragraph beginning on page 12, we discuss how fractionated ESRRB-expressing populations compare to EpiSCs, EpiLCs and epiblast cells from post-implantation embryos at distinct developmental stages.

5. In the comparison between the naïve and primed state of pluripotency, how do the authors explain about the fact of the Nanog expression in both states? Is it reactivated in the primed state?

Yes, Nanog is reactivated in the primed state. RNA in situ expression in the embryo (Chambers et al., 2003) and unpublished data from our lab examining Nanog protein expression shows that Nanog expression is extinguished in the late blastocyst. Expression analyses of RNA (Chambers et al., 2003; Hart et al., 2004), expression of a GFP reporter placed at the Nanog locus (Osorno et al., 2012) and whole mount immunofluorescence (Chambers lab, unpublished; Hoffman et al. 2013) shows that Nanog is re-expressed in the post-implantation epiblast. It is known that Esrrb is not expressed in the post-implantation epiblast (Adachi Mol. Cell 2013).

Differences in the nuclear and/or signalling environment of primed and naïve pluripotent cells are likely to be responsible for re-expression of Nanog but not Esrrb. We have added text dealing with these points on pages 22-23. These will be interesting and important avenues for future studies.

6. Page 6; The authors proposed that Nanog positions at upstream of Esrrb based on the over-expression data. How about their link in knockout ES cells? How could this idea be consistent with the presence of Nanog-negative; Esrrb-positive cells?

The proposition that Nanog acts upstream of Esrrb is not solely dependent upon these overexpression analyses but builds upon our previous work (Festuccia et al., 2012). In brief, we showed by relocalising a Nanog-ERT2 fusion protein to the nucleus of Nanog-null ES cells using tamoxifen, that Esrrb is the most prominently upregulated gene that responds to Nanog. By quantitative RT-PCR of intronic (therefore, pre-mRNA) we could detect an increase in Nanog transcription within minutes of Tamoxifen application. These prior studies unequivocally establish that the Nanog protein acts upstream of Esrrb gene. In a separate study (van den Berg, Mol Cell Biol, 2008), we provided evidence that DNA sequences within the proximal Nanog promoter that coincided with the position of an Esrrb-binding site (measured by EMSA) had a positive effect on the activity of a luciferase reporter in ES cells. The experiments in Figure 1 were therefore performed to assess more fully the ability of each of these proteins to alter the expression of the other in ES cells. We realise now that we have not clearly recapitulated these previous findings on the potential regulation of the Esrrb gene by NANOG and of the Nanog gene by ESRRB. Therefore, we have re-written the first paragraph of the section entitled 'The epistatic...' to make previous knowledge clearer and to therefore put in context the fact that what we were testing was not so much whether Nanog acts upstream of Esrrb, which has been conclusively demonstrated (Festuccia et al. 2012) but the extent to which previous luciferase based assays implicating Esrrb binding to the Nanog promoter actually meant that Esrrb had a demonstrable effect on Nanog expression at the endogenous loci. By Esrrb overexpression we show that Esrrb does not drive Nanog expression. Therefore the simple relationship between these 2 genes can be clearly stated as Nanog → Esrrb but not the opposite. Importantly, we recognise that this does not exclude additional inputs into the regulation of the 2 genes. Indeed, we never claim that Nanog is the sole activator of Esrrb expression in ESCs; the context appears to be important too. This most likely explains why in the post-implantation epiblast (and in EpiSCs), Nanog is re-expressed but Esrrb is not.

Data from Nanog knockout ES cell lines also support our conclusions. As shown in Fig S4B, in Nanog KO ES cells, Esrrb levels are generally attenuated compared to WT lines which show an identical reporter configuration. Most importantly, the proportion of Esrrb negative cells is greatly increased. Reciprocal data from Esrrb KO ES cells would be unfortunately difficult to interpret in this respect. Esrrb KO ES cells kept in FCS/LIF show extensive spontaneous differentiation. In this context any fine effect of Esrrb on Nanog expression would be masked by loss of Nanog expression in differentiated cells.

As for the presence of an Esrrb-positive fraction in Nanog-negative ES cells, we propose there is a temporal delay between Nanog downregulation and Esrrb downregulation. Loss of Nanog does not appear to cause an immediate loss of Esrrb expression. Rather, ES cells remain in a transient state in which the complete silencing of Esrrb expression is more likely to occur (compare left and right bottom panels in Fig 2B: an Esrrb negative population appears only at day 3, in cells that have reduced Nanog expression). Potentially, in the absence of Nanog other positive inputs onto Esrrb are able to temporarily maintain Esrrb expression, but these are ultimately destined to fail when not re-inforced by Nanog.

7. Page 8; The authors stated that they analyzed the SSEA-1+ cells. Did SSEA-1+ selection cover all Nanog-positive or Esrrb-positive subpopulations? Is any bias ruled out here?

We have compared the double reporter profiles after selecting the highest and lowest 30% of the SSEA1-stained population. As can be seen in the panel below there is no obvious bias introduced by selecting SSEA1 positive cells for our analyses. In particular, analysis of SSEA-low cells provides no evidence of a previously overlooked Nanog⁺, Esrrb⁻ population in SSEA1 low cells.

Figure:

8. Page 9; The authors proposed that Nanog acts upstream of Esrrb and Klf4 since the selection for Nanog expression resulted in their homogeneous expression. However, this could also be explained by parallel position or Nanog positioning downstream of Esrrb and Klf4.

We have shown previously (Festuccia et al., 2012) that Klf4 responds quickly to induction of Nanog. Given this and the previously discussed findings that Esrrb is a direct target of Nanog, we consider that our interpretation of the effects of selecting for Nanog expression is valid. In the submitted version of the manuscript there was a typographical error which removed “Klf4 is also a primary direct target of..” from the start of the paragraph. We apologise for this error, which has been rectified. Importantly, the reviewer points out that additional inputs onto Nanog target genes exist. We acknowledge this and have amended the text of the paragraph to:

“Our previous study has shown that Klf4 is also a direct primary target of NANOG (Festuccia, et al. 2012). Analysis of KLF4 expression by intracellular cytometry in TNG E-tdT cells suggests that ESCs downregulate KLF4 at a similar rate to Esrrb (Fig. S5A). Selection for Nanog expression using puromycin abolished heterogeneous expression of both Esrrb-tdTomato and KLF4 (Fig. S5B). While this result is compatible with placement of Nanog upstream of Klf4 as well as Esrrb, we note that additional positive inputs into Klf4 exist (Niwa et al. 2009).”

9. Page 11; The authors emphasized the close relation between Esrrb-negative cells and EpiSCs. Is it possible to make EpiSC-like cells from Esrrb-negative cells easier than the unselected ES cells in vitro?

To further investigate the relationship between these cells, we have compared mRNA expression in sorted sub-populations of Esrrb:dGFP1 cells (Fig. 3G). This shows that Esrrb^{negative} cells are transcriptionally more similar to EpiLCs than to ESCs or EpiSCs. To address the reviewer’s point we tested fractionated cells for gene expression after replating in EpiSC differentiation media (Fig. S5B). This shows that Esrrb^{negative} cells progress in early differentiation stages faster than Esrrb^{high} cells.

10. Page 12; 'upregulated during the Esrrb-GFP negative to EpiSC transition'---this is confusing statement due to the lack of evidence for direct transition.

We have corrected the wording to state 'upregulated in EpiSCs compared to Esrrb-negative cells'.

11. In relation to Figure 5, how about the binding of Oct4 and Nanog in EpiSCs in their relationships to the patterns observed in Esrrb-high and low cells?

We discuss how our results relate to published reports describing the binding of Oct4 in primed cells in Fig 6B; in these cells Nanog binding could not be assessed since Nanog is not expressed in EpiLCs. The reviewer asks where Nanog might bind when it is re-expressed in EpiSCs. This is an interesting point that is being addressed in other projects in the lab. However, these studies are ongoing and do not directly impinge upon this study which is focussed on the transition out of naïve pluripotency. We hope the reviewer will agree that a proper consideration of the binding of these TFs in EpiSCs is beyond the scope of the current study.

12. Page 17; The authors described that the motif analysis with RSAT resulted in over-representation of ESRRB, KLF4 and KLF5 binding sites at Class I peaks and Oct4 binding site at Class II peaks. I am not familiar with RSAT analysis, and I wonder how these results can be interpreted. Do they the only represent sequences over-represented to oct4 binding sites of each class? Why not Oct-Sox motifs? Are there any motifs over-represented nearby (~100 bp) the Class II peaks?

The RSAT software is a tool for de-novo motif discovery similar to the popular MEME suite, but more appropriate for larger datasets such as peak lists from ChIP-seq experiments. In brief, RSAT looks for short recurrent patterns ("motifs") in a set of DNA sequences that might represent the potential binding sites of transcription factors in given regions. RSAT compares any discovered motifs to databases of binding motifs that have been linked to specific binding proteins by other experiments reported in the literature, so that one can easily make sense of the discovered patterns. Here we provided RSAT with the DNA sequences underlying class I and class II peaks. We then ran the motif discovery in differential mode, i.e., we asked for DNA motifs that occurred preferentially in one class but not the other. Matching the discovered motifs to the JASPAR database, showed enrichment for Esrrb and Klf4 in class I peaks and Oct4 in class II peaks. The presence of these motifs in the peak groups provides further supporting evidence for the potential regulatory role of these TFs at the given enhancers.

The Oct-Sox motif (JASPAR database MA0142.1: http://jaspar2014.genereg.net/cgi-bin/jaspar_db.pl?ID=MA0142.1&rm=present&collection=CORE) occurs with similar frequency in both peak groups and was therefore not reported by the discriminatory analysis (we now mention this in the text on page 17). This was established using the matrix-scan tool of the RSAT software (http://rsat01.biologie.ens.fr/rsa-tools/matrix-scan-quick_form.cgi) to scan the peak sequences of both groups specifically for the Oct-Sox motif. Indeed, we observed only a small difference in hit frequency (sites matching motif at $p \leq 1e-4$: class I = 1,097 (54%), n(class II) = 1,133 (72%)), indicating that Sox2 colocalization with Oct4 does not distinguish between the enhancer classes. The reviewer raises an interesting point about sequences represented in the proximity of class II peaks. We understand the reviewer is asking whether their might be any other factors bound in the close proximity of these peaks that could be critical to the observed behaviour. To answer this question more thoroughly, we repeated the motif analysis looking at a broader window around the peak centre (+/-400bp instead of +/-200bp used previously) that therefore includes neighbouring DNA fragments. Since the web server version of the RSAT software had been updated since we last used it, we first repeated the original analyses (+/-200bp) at this point in order to keep all results consistent. Reassuringly, all our previously obtained results still held up as reported. No new motifs were discovered when we performed the same analysis (with +/-400bp window) on class I vs. class II peaks. In this case, the motifs discovered previously (Esrrb-like and Klf-like) were again the top hits and the only ones that were significant.

We have now edited the text to make it more explicit that we report the results of a differential motif analysis (page 17) and we also report significance metrics (Fig 6C). However, as neither the repeat analysis with the updated software version nor the analysis with the extended window size revealed any new insights, we have not altered the results presented in the initial submission.

13. For the regulation of Nanog and Esrrb by extrinsic signals, it was reported that Nanog expression is sensitive to the MAPK signal whereas Esrrb is preferentially activated by Gsk3

inhibition, suggesting their differential regulation. Although the authors argued about the hierarchy between them, it will be possible that they are just in parallel or non-hierarchical relation.

This comment relates to a previous point (#6) raised by the reviewer. As we have discussed above, the notion that Esrrb is a downstream target of Nanog is supported by a combination of published data and evidence we present in this manuscript, including overexpression in WT cells (Fig 1) and gain of Nanog function in KO reporter lines (Fig S4). As also noted above, we do not suggest that Nanog is the sole activator of Esrrb. The reports mentioned by the reviewer are therefore not in conflict with our discussion of the results. It is possible that GSK3 inhibition acts in parallel to Nanog to promote Esrrb expression. Unpublished data generated in the lab shows that Esrrb expression becomes homogeneous in FCS/LIF cultures treated with MEK inhibitors, raising the possibility that MAPK signalling also control Esrrb expression, possibly through Nanog. Although important, we feel that these observations are not directly pertinent to the focus of the present manuscript.

Referee #2:

Festuccia et al report on the differences in Esrrb expression states in mouse ES cells, and their relationship with loss of naïve pluripotency. The authors document that Esrrb defines three different populations within ESC cultures under serum/LIF conditions, based on the expression levels of Esrrb. The negative Esrrb population is proposed as the population which first exits naïve pluripotency. This conclusion is supported by a number of experiments based on reporter, and endogenous expression of Nanog and Esrrb, as well as by colony formation and AP analysis. In addition, the authors show that the enhancers of Esrrb and Nanog show indeed gradual changes in the Esrrb-high, low, and negative, suggesting that the Esrrb-low are indeed of intermediate nature upon pluripotency exit.

The data presented is carefully controlled and the manuscript for the most part well written according to the data presented. The manuscript is novel, and presents new insights in the understanding of transitional states upon exit from pluripotency. It identifies Esrrb expression as a marker of the earliest step upon exit and therefore as the 'first gatekeeper' of naïve pluripotency. These are all new data that will have a big impact on the stem cell community. I am therefore fully supportive for publication in EMBO J, provided the authors address my following points. *We thank this reviewer for their enthusiastic support of our work.*

1. On page 7, related to Figure 2. Because Nanog-low and Esrrb-low cells are within the same population, it is impossible to disentangle who is doing what (e.g. which population 'precedes' which population). Towards this end, can the authors perform the same analysis as in Figure 2A (daily analysis of fluorescence) of Nanog expression in Esrrb-/- cells and viceversa (probably an inducible shRNA or siRNA approach may be needed)? The data presented in the supplementary Figure S4 provides a partial answer to this issue, but based on this data, I disagree with the authors in their conclusions that 'loss of Esrrb expression was uncoupled from down regulation of the activity of the Nanog locus' (Page9).

We agree with the reviewer that the analysis performed on Nanog null double reporter lines provides useful insights into the order in which Nanog and Esrrb are downregulated. The previous FigS4 (now Fig 1H) shows the Nanog and Esrrb expression in Nanog:GFP/Esrrb:TdTomato ES cells, and compares the profiles detected in lines expressing (TNG) or devoid (ESDN) of Nanog. In these experiments cells cultured at high density are plated for three days at relatively low density in FCS/LIF. These conditions are the same as those used in Fig1 and Fig2 (Day 3 data specifically). After release for 3 days from conditions in which heterogeneity in Esrrb and Nanog expression is minimal, cells lacking a functional Nanog allele (ESDN), downregulate Esrrb to a greater extent and in a higher fraction of the population compared to cells with an intact Nanog allele (Esrrb-low/negative cells are 62.5% of the total population in ESDN cells versus 35.1% in TNG cells). This suggests that absence of NANOG favours loss of Esrrb. Coupled to the notion that NANOG-high cells cannot downregulate Esrrb, this strongly favours the view that in WT cells Nanog downregulation must precede the appearance of an Esrrb negative population. Conversely, in Nanog null double reporter lines it is possible to detect Esrrb downregulation in cells showing high transcriptional activity from the Nanog:GFP allele - which in these cells cannot result in accumulation of any NANOG protein.

Incidentally, the latter is the idea that the sentence "loss of Esrrb expression was uncoupled from down regulation of the activity of the Nanog locus" was intended to convey. Nonetheless, we agree

with the reviewer that a mild inverse correlation between Nanog:GFP expression levels and Esrrb downregulation persists in Nanog null reporter lines. To amend this imprecision we have now changed the statement criticised by the reviewer to: "loss of Esrrb expression was not necessarily coupled to downregulation of the activity of the Nanog locus".

We also agree with the reviewer that performing reciprocal experiments in Esrrb-null reporter lines would be instructive. After considering this option, we concluded that unfortunately such experiments are unlikely to provide interpretable results due to the extremely unstable self-renewal of Esrrb-null ESCs. Any fine effect of Esrrb on the kinetics of Nanog downregulation would be masked by a preponderant fraction of differentiating or differentiated cells in the cultures, making it impossible to distinguish direct from indirect effects.

Given these limitations, and since the point raised by the reviewer is of importance, we decided to address the problem more directly and performed experiments to further characterise the sequence of events leading to the appearance of Esrrb-negative/Nanog-negative cells. We have added to Fig2 the results of an immunofluorescence timecourse analysis of the downregulation of Nanog and Esrrb upon exit from 2i culture. After an initial drop in expression of both Nanog and Esrrb (day 1), due to the removal of inhibitors, cells reduce Nanog expression further, with a substantial fraction of the population close to/ or below the negative expression threshold. Esrrb levels tend to follow the changes in Nanog expression closely, but the levels of Esrrb remain clearly above the negative threshold (set on RA differentiated cells) at day 2. At day 3, Esrrb expression levels drop under the negative threshold in a fraction of the Nanog low population. From these dynamics, and from the observation that it is possible to detect cells with low Nanog expression and relatively high Esrrb but not vice-versa (see shape of the trendline in Fig1B), we conclude that Nanog downregulation precedes Esrrb downregulation. We propose that downregulation of Nanog below a certain threshold places ES cells in a transient state in which the complete silencing of Esrrb expression has a higher probability to occur. However, since the Esrrb-high/Nanog-low sub-population is less prominent after staining WT cells, we conclude that less emphasis should be placed in claiming the existence of a STABLE subpopulation of cells that have completely downregulated Nanog and still shows elevated Esrrb expression. We have therefore re-written sections of the paper accordingly and have reduced our interpretation of the experiments previously reported in Fig. 2. We hope the reviewer considers that our revised manuscript satisfactorily addresses the concern that he/she has raised.

2. Related to Figure 5, on page 16: the authors conclude that "Nanog displayed a marked reduction in binding to its targets globally in Esrrb-neg cells". This is unsurprising, considering that NANOG protein is practically gone in these cells (as shown in the WB on Figure 4C). It is also unsurprising that ChIP for Nanog and Esrrb shows a reduction in their binding in the ChIP presented in Figure 4D in the Esrrb-negative cells, so the extent of the relevance of a reduction in binding here is questionable, since it is basically the expression that goes down, not the binding (e.g. the binding is not regulated, their expression is).

We agree with the reviewer that, given the reduced Nanog protein levels in Esrrb-negative cells, the loss of Nanog and Esrrb binding is unsurprising. However, the more interesting point is the differential binding of Oct4 at class I and class II elements. We have therefore reduced the text on the relevant section (page 15) to help the reader focus on this more interesting aspect.

In contrast, the analysis of H3K27ac is much stronger and rather interesting. Can the authors complement this ChIP with additional enhancer and promoter marks (e.g. H3K4me1, H3K122ac and/or H3K4me3)?

As suggested, we have performed additional histone modification analyses (Fig. 4 and S7). These show that in Esrrb-negative cells H3K4me1 and H3K4me3 are lost from the Esrrb enhancer (Class I example) but remain detectable at the Nanog enhancer (Class II example). In light of the recent debate on the mechanistic function that these marks might or might not play in controlling distal regulatory elements, we believe that these additional data constitute a valuable example in which chromatin modification changes relate to variations in enhancer function. We briefly comment on these new findings and put them in the context of the recent literature in the discussion section at page 25. Unfortunately, we were unable to get any meaningful data on H3K122ac, despite 2 attempts using commercial antibodies.

3. Same applies for the genes that loose Nanog binding in Figure 5D and G. How many peaks 'gained' Nanog binding in Figure 5D, are these numbers really relevant?

The number of peaks gained is indicated in Fig. 5C as 49. We agree that this low number probably reflects low biological significance. The plots shown in Fig. 5D indicate that all these sites already

had a low (but sub-threshold) level of binding in *Esrrb*(high) and *Esrrb*(medium) cells, with <2-fold change between *Esrrb*-medium and *Esrrb*-negative. The phenomenon of “gained Nanog peaks” is therefore likely to be a consequence of the chosen analysis thresholds. Therefore, we have minimised discussion of Nanog ‘gained’ peaks in the text. Nevertheless, in the interests of transparency, we have left Figure 5 unchanged. Overall, this has no impact on the conclusions presented in this manuscript but helps to frame the effects detected on the 9,188 sites that lose Nanog binding (and that we believe are biologically relevant). These sites in fact display a much stronger change of signal intensity compared to gained peaks (see centre panel in Fig. 5D).

Minor comments.

1. Some sentences are a little overstated, so please rephrase (example, page 4, ‘...enabled transcriptional and epigenetic changes accompanying extinction of the naïve pluripotency network to be identified’): the authors only looked at changes in DNA methylation by restriction enzyme/PCR, and H3K27ac was only looked at on the Nanog and *Esrrb* genes), no global ‘epigenetic’ analysis is performed thoroughly. Another example on page 9, first two lines of the third paragraph).

We have modified the wording as the reviewer suggests to make less overblown claims on our data. We have checked the rest of the text to reduce additional such examples.

2. The nomenclature of H3K27Ac and H3K27(me)₃ is incorrect (it should be H3K27ac and H3K27me₃).

These changes have been made as suggested.

3. Throughout the figures. It would be beneficial to add small ‘diagrammes’ of the reporters used in the panels, as well as the genotype of the cell lines in the figures (e.g. Nanog +/-, *Esrrb* - wt, etc.): the nomenclature EF4, E14Tg2a, TNG ...etc is well known by only few people. Considering the broad readership of EMBO J., the authors could make an effort to make the data more accessible to a wider community.

We draw the reviewer’s attention to Figure S1 which has diagrams of all the cell lines we use in the study. In each of the figures / legends we have added a note pointing to the relevant panel of Figure S1. We respectfully consider that this is sufficient and that additional diagrams in each figure would be unnecessarily obtrusive. In addition, we have replaced instances of arcane nomenclature as the reviewer has suggested.

4. Figure S3 is not numbered in order in the manuscript.

We have re-checked the numbering of all figures in the manuscript. Notably, we found that Figure S7 B and C were called before Figure S7A. This has been corrected in the current version.

5. The identification of the DNA binding motifs in Figure 6C is not meaningful unless p-values are presented.

We thank the reviewer for pointing out this lack of detail. We now report the significance metrics output by RSAT (see Fig. 6C and our response to question #12 of reviewer 1).

For each motif, RSAT reports the enrichment of the most significant intermediate motif that was assembled into the final consensus motif. For the three motifs shown in Figure 6C these values are:

Class I:

**Esrrb*: $p = 8.6e-28$*

**Klf1/4/5*: $p = 3.4e-14$*

Class II:

**Oct4*: $p = 8.6e-20$*

The three reported motifs were the only ones considered significant at a threshold of ≥ 9 on the significance score (e-value $\leq 1e-9$).

6. Data presented in Figure S4 is quite important, it would be best to include it in the main manuscript.

We agree with the reviewer that the data presented in Figure S4 is important. For this reason we moved it to Fig 1H. We also devote a full paragraph to discuss the relevance of these findings on page 8, minimising the possibility that these results are overlooked by the reader.

2nd Editorial Decision

13th Jul 2018

Thank you for submitting your revised manuscript for consideration by The EMBO Journal. Your revised study was sent back to referees #1 and #3 for re-evaluation, and we have received comments from both of them, which I enclose below. As you will see the referees find that their concerns have been sufficiently addressed and they are now broadly favour of publication.

Thus, we are pleased to inform you that your manuscript has been accepted in principle for publication in The EMBO Journal, pending some minor issues regarding material and methods, data representation and formatting as outlined below, which need to be adjusted at re-submission.

REFEREE COMMENTS:

Referee #1:

In this revised manuscript, the authors addressed the points raised by the reviewers and gave clear answers. Now this reviewer is satisfied and agrees with the publication of the present version of the manuscript.

Referee #3:

The authors have made a significant effort to address the points that I raised during my original review. While some points were not technically feasible, the manuscript is improved and in my view, suitable for publication as is, considering that the overall message of the paper is novel and will bring important new insights to our understanding of gene regulatory networks in stem cells. I have only a very minor 'picky' item, which is not intended to delay acceptance, but which the authors may want to revise in their final edited manuscript: wouldn't it be most appropriate to show a non-cropped WB in Figure S3A?

Corresponding Author Name: Ian Chambers

Journal Submitted to: EMBO J

Manuscript Number: EMBOJ-2016-95476R